

# Conservation tillage and fertiliser management strategies impact on basmati rice (*Oryza sativa* L): crop performance, crop water productivity, nutrient uptake and fertility status of the soil under rice-wheat cropping system

Mandapelli Sharath Chandra[1], R.K. Naresh[1], Rajan Bhatt[2], Praveen V. Kadam[3], Manzer H. Siddiqui[4], Abdel-Rhman Z. Gaafar[4] and Md Atikur Rahman[5]

[1] Sardar Vallabhbhai Patel University of Agriculture & Technology, Meerut, India
[2] Punjab Agricultural University, Kapurthala, India
[3] Indian Agricultural Research Institute, New Delhi, India
[4] Department of Botany and Microbiology, King Saud University, Riyadh, Saudi Arabia
[5] National Institute of Animal Science, Rural Development Administration, Cheonan, Republic of Korea

Corresponding author
Manzer H. Siddiqui,
mhsiddiqui@ksu.edu.sa

## ABSTRACT

**Background**. The sustainability of paddy production systems in South Asia has recently been affected by a decline in soil health and excessive water usage. As a response to the global energy crisis, escalating costs of synthetic fertilisers, and growing environmental concerns, the utilization of organic plant-nutrient sources has gained considerable attention. Emerging adaptation technologies, including conservation tillage and innovative approaches to fertilizer management, present practical choices that can significantly contribute to the long-term preservation of soil fertility.

**Methods**. The two year-long field experiment was completed in sandy loam soil during rainy (*Kharif*) seasons in 2019 and 2020 at the crop research centre farm of Sardar Vallabhbhai Patel University of Agricultural & Technology, Meerut, Uttar Pradesh to analyze the impacts of different tillage establishment of the crop and its methodologies as well as integrated nutritional management approaches on rice growth, yield, productivity of water, nutrient uptake, and fertility status of soil under a rice-wheat rotation system. The experiment was set up in a factorial randomized block design and replicated three times in a semi-arid subtropical environment.

**Results**. The conventionally transplanted rice puddled (CT-TPR) grew substantially better taller plants, and higher dry matter buildup leads to increased yields than transplanted rice under raised wide bed (WBed-TPR). WBed-TPR plots had more tillers, LAI, CGR, RGR, and yield characteristics of the rice in two year study. CT-TPR increased grain yield by 4.39 and 4.03% over WBed-TPR in 2019 and 2020, while WBed-TPR produced the highest water productivity ($0.44 \text{ kg m}^{-3}$) than CT-TPR, respectively. The 100% RDF+ ZnSO4 25 kg ha$^{-1}$ + FYM (5 t ha$^{-1}$) + PSB (5 kg ha$^{-1}$) + Azotobacter 20 kg ha$^{-1}$ (N6) treatment outperformed the other fertiliser management practices in terms of crop growth parameters, yields of grain (4,903 and 5,018 kg ha$^{-1}$), nutrient uptake and NPK availability, organic soil carbon. Among the fertilizer management practices, with the direct applications of the recommended dose of fertilizer (RDF),

farm yard manure (FYM), phosphate solubilizing bacteria (PSB), Azatobactor and zinc worked synergistically and increased grain yields by 53.4, 51.3, 47.9 and 46.2% over their respective control treatment.

**Conclusions**. To enhance rice productivity and promote soil health, the study suggests that adopting conservation tillage-based establishment practices and implementing effective fertilizer management techniques could serve as practical alternatives. It is concluded that the rice yield was improved by the inclusive use of inorganic fertiliser and organic manure (FYM). Additionally, the study observed that the combination of conventional puddled transplanted rice (CT-TPR) and N6 nitrogen application resulted in enhanced rice crop productivity and improved soil health.

# INTRODUCTION

Rice, commonly known as paddy, is a critical crop that supplies 19% of the world's nutritional energy and serves as a staple food for about half of the global population (*Tyagi et al., 2022*). To meet India's food needs by 2050, it is projected that food supply will need to increase by 60% (*Tyagi et al., 2022*). Rice uses 27% of global freshwater (*Bouman, Lampayan & Tuong, 2007*). Puddles receive 30% of wetland rice irrigation. Thus, Asia's irrigated rice fields will run out of water by 2025, necessitating water conservation (*Md Alam et al., 2020*). Rice and wheat crop-establishing strategies (CETs) and management are being prioritized (*Shahane et al., 2020*). CETs vary in resource consumption, energy needs, and climate change mitigation, which can affect farmers' produce, income, and environmental health. One of them, India, transplants seedlings into puddle soil by hand (*Nahar et al., 2017*). Continuous conventional puddled rice transplanting diminishes water and land productivity, degrades soil structure, and lowers subsurface water levels. New CETs and fertiliser management techniques are needed to address environmental resource depletion and escalating synthetic and agronomic costs (*Shahane et al., 2020*). Zero cultivation, dry direct sowing, wet sowing, water spawning, strip sowing, bed revegetation, non-puddled rice transplanting, mechanised rice transplantation, and combinations thereof have been developed to reduce these negative effects. These strategies may reduce global warming, resource conservation, crop production, soil health, and other issues (*Md Alam et al., 2020*; *Drechsel et al., 2015*).

Fertilizer is essential for agricultural production. Farmers often use excessive fertiliser per crop without considering the specific nutrient requirements of the crop. This practice leads to an imbalance of nutrients in the soil subsequently resulting in decreased crop yields. Due to centuries of continuous agriculture, the utilization of modern agricultural equipment, and inadequate fertilizer application practices, the unbalanced use of inorganic fertilizers has led to a decline in soil fertility. Consequently, Indian soils, in general, are characterized by a state of infertility (*Mahmud, Shamsuddoha & Haque, 2016*). The ongoing decline

in soil health can be attributed to the incorrect application of synthetic fertilizers and the insufficient recycling of organic waste (*Nahar et al., 2017*). Since the introduction of synthetic fertilizers, rice farmers have increasingly relied on inorganic fertilizers, leading to their excessive use. However, data show that the recurrent use of chemical fertilisers alone degrades soil physical characteristics and organic matter levels. As a result, this approach fails to sustain desired output and poses significant harm to soil health (*Mohammad, 2010*). Therefore, adopting a judicious approach and applying inorganic fertilizers sparingly can be beneficial in increasing rice yield (*Haque & Haque, 2016*). Organic nutrients support beneficial microorganisms, enzymes, and soil physical and chemical properties, which help sustain soil health. Chemical fertiliser boosts microbial activity, nutrient uptake, and plant nutrient availability when applied with organic manure. Combining organic and chemical fertilisers offers advantages (*Roba, 2018*). Thus, organic manure and inorganic fertilisers boost cereal performance by retaining yield and soil health (*Mahmud, Shamsuddoha & Haque, 2016*; *Sharma et al., 2017*). The current study at our university campus examined the effects of tillage techniques and INM options on rice growth, yield characteristics, yield, water productivity, and post-harvest nutrient status in *typic ustochrept* soils of northwestern U.P., India.

## MATERIALS & METHODS

### Place of the research
The field experiment in the *Kharif* seasons 2019–20 was conducted at the CRC Farm, College of Agriculture, SVP University of Agriculture & Technology, located near Meerut on the Indo-Gangetic Plains of Western Uttar Pradesh, India. The farm is 232 m above sea level, 29°08′12′N latitude, and 77°40′ 52′E longitude.

### Weather and climate
A local meteorological observatory measured temperature, humidity, sunlight hours, rainfall, and wind speed daily throughout the experiment. The study site is characterized as semi-arid to subtropical and receiving an average annual rainfall of 845 mm. The majority of this rainfall, around 80–90%, occurs between the months of June and September. Frost usually appears in December and lasts until January throughout winter, which lasts from November to February. In May, the average temperature is 43–45 °C, while in winter it is 3 °C. Figure 1 show that across the crop period in 2019 and 2020, the average maximum and minimum weekly temperatures ranged from 40.3 and 38.5 °C to 15.9 and 16.1 °C. The mean maximum and minimum relative humidity dropped from 95.8 and 88.1 to 46.8 and 41.7% over the crop period in 2019 and 2020. In the first and second years of research, total rainfall was 587.6 and 369.8 mm, respectively.

### Organization and experimentation
Three factorial randomised block designs reproduced the experiment. Table 1 shows its treatments: two rice planting methods and nine fertiliser management methods. The experimental plot was 10.0 m × 3.0 m. Pusa Basmati 1509 (PB 1509) is a high yielding, premium quality, semi-dwarf (115–120 cm), climate smart, water-saving, fertiliser

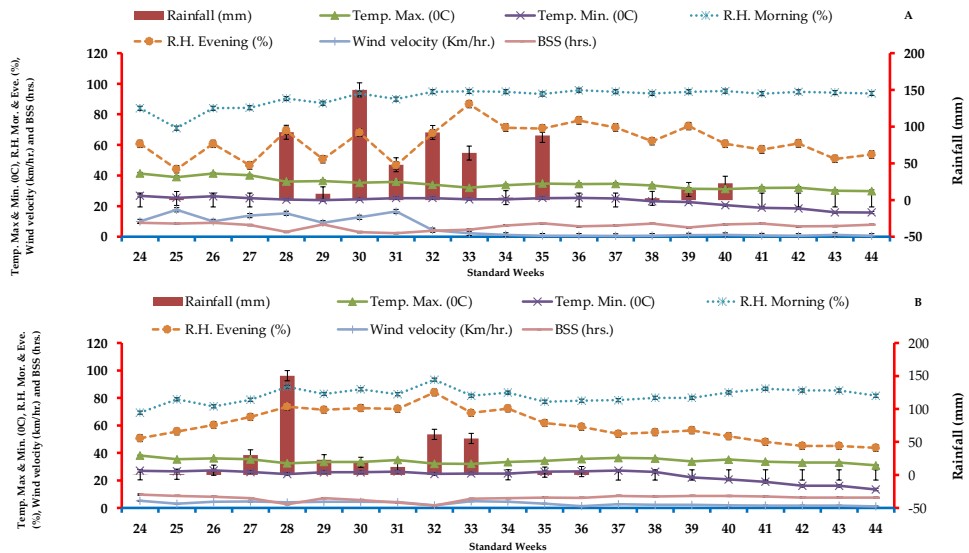

**Figure 1** Data on agrometeorological averaged weekly throughout the crop duration *kharif* season in 2019 (A) and (B) 2020 (B) at experimentation location.

responsive, short duration, and semi-dwarf basmati rice variety developed at ICAR-Indian Agricultural Research Institute, New Delhi in 2013. Brown spot and leaf blast diseases are resistant to it to a modest extent. The grain length after cooking is very good (18 to 19 mm), the ASV is desirable (7.0), the amylose content is intermediate (21 to 22%), and the aroma is powerful. The grain length is extra-long and thin (8 to 9 mm) with very infrequent grain chalkiness. The National Capital Region, Haryan, Delhi, and Punjab are suggested locations for basmati cultivation on a huge scale. At 21 days old, PB-1509 rice seedlings were transplanted at 25 cm by 10 cm or 20 cm by 10 cm in the transplanted wide bed method. Bunds marked experimental field plots with proper irrigation channels.

## Weed management

The plots remained weed-free throughout the season. Bispyribac sodium (Nominee gold) and butachlor @25 g a.i. ha$^{-1}$ and 1,300 g a.i. ha$^{-1}$ was sprayed one month after transplantation. Single-hand weeding keeps transplanted rice patches weed-free.

## Application of fertilizer and soil characteristics

The examined soil was sandy loam, non-saline, moderately alkaline, low in organic carbon, and low in accessible nitrogen (235.8 and 242.5 kg ha$^{-1}$), phosphorus (18.6 and 19.8 kg ha$^{-1}$), and potassium. (K$_2$O- 210.5 and 215.6 kg ha$^{-1}$). Table 2 shows 2019–20 growth season parameters. The appropriate fertiliser amounts of 150, 60, and 40 kg N, P$_2$O$_5$, and K$_2$O ha$^{-1}$ were applied. Urea, SSP, and muriate of potash will apply the right amounts of N, P, and K. Before the field's last plough, full P, K, and half N doses were utilised as a baseline regimen. Rice was split-applied 50% N during active tillering and panicle commencement. 5 kg ha$^{-1}$ PSB, 20 kg ha$^{-1}$ *Azotobacter*, 0.5% N, 0.2% P, and 0.5% K FYM were soil sprayed

**Table 1  Details of the treatments.**

| Treatments | |
|---|---|
| **Crop establishment (CE) methods:** | |
| $CE_1$ | Conventionally tillage transplanted puddled rice (CT- TPR) |
| $CE_2$ | Conservation tillage wide bed Transplanted rice (W Bed-TPR) |
| **Fertiliser management strategies:** | |
| $N_1$ | Control |
| $N_2$ | 100% RDF (150:60:40:: $N:P_2O_5:K_2O$ kg ha$^{-1}$) + $ZnSO_4$ 25 kg ha$^{-1}$ |
| $N_3$ | 125% RDN (187.5:60:40:: $N:P_2O_5:K_2O$ kg ha$^{-1}$) + $ZnSO_4$ 25 kg ha$^{-1}$ |
| $N_4$ | STCR based NPK application + $ZnSO_4$ 25 kg ha$^{-1}$ |
| $N_5$ | 100% RDF + FYM (5 t ha$^{-1}$) + $ZnSO_4$ 25 kg ha$^{-1}$ |
| $N_6$ | 100% RDF + FYM (5 t ha$^{-1}$) + PSB (5 kg ha$^{-1}$)+ Azotobactor 20 kg ha$^{-1}$+ $ZnSO_4$ 25 kg ha$^{-1}$ |
| $N_7$ | 75% RDN (112.5: N kg ha$^{-1}$)+ FYM (5 t ha$^{-1}$) + $ZnSO_4$ 25 kg ha$^{-1}$ |
| $N_8$ | 75% RDN (112.5: N kg ha$^{-1}$)+ FYM (5 t ha$^{-1}$) + PSB (5 kg ha$^{-1}$) + Azotobactor 20 kg ha$^{-1}$+ $ZnSO_4$ 25 kg ha$^{-1}$ |
| $N_9$ | Organics Practices @ FYM (30 t ha$^{-1}$)+PSB (5 kg ha$^{-1}$) + Azotobactor 20 kg ha$^{-1}$ + $ZnSO_4$ 25 kg ha$^{-1}$ |

one week after rice transplantation. IARI's STCR formula is 6.97 X T - 0.38 X SN, 5.73 X T - 4.81 X SP, and 3.92 X T - 0.28 X SK (*Argal, 2017*).

## Water studies
### Application of water and its measurements
A water metre measured water applied to each plot through 15-cm poly vinyl pipes in the irrigation system. (Dasmesh Co., Punjab, India). The formulae will calculate irrigation depths and water supply:

$$\text{Applied water quantity (L)} = F \times t \tag{1}$$

$$\text{Applied water depth (mm)} = L/A/1000. \tag{2}$$

Thus, F denotes flow (L/s), t indicates irrigation time (s), and A indicates plot area. (m2). A meteorological station rain gauge will record rainfall. Irrigation and rainfall were added to calculate water supply (input water). The formula for water productivity (WPI+R) (kg/m3) is *Humphreys et al. (2006)*.

$$WP_{I+R} = \text{Grain productivity}/(\text{Applied irrigation water}(I) + \text{Crop received rainfall}(R)). \tag{3}$$

### Water productivity
Irrigation water productivity (WP) is calculated by dividing crop output by water utilised (WPIRRI), total crop water demand or gross inflow (WPTCW), and evapotranspiration

**Table 2 Initial status of the soil in experimental field (Kharif 2019 & 2020).**

| Particulars | Values | | Approach to analysis |
|---|---|---|---|
| | **2019** | **2020** | |
| **Physical characteristics** | | | |
| **Mechanical analysis** | | | |
| Sand (%) | 45.50 | 46.82 | |
| Silt (%) | 24.60 | 22.98 | Bouyoucos hydrometer *Piper (1966)* |
| Clay (%) | 29.90 | 30.20 | |
| Textural class | Sandy loam | | |
| **Physico-chemical characteristics** | | | |
| Soil pH | 7.9 | 8.2 | Digital pH meter DI 707 *Jackson (1967)* |
| Electrical conductivity (dsm$^{-1}$) | 0.32 | 0.28 | Digital EC meter *Jackson (1967)* |
| **Chemical characteristics** | | | |
| Organic carbon (%) | 0.49 | 0.52 | Wet digestion method *Walkley & Black (1934)* |
| Available nitrogen (kg ha$^{-1}$) | 235.8 | 242.5 | Alkaline permanganate method *Subbiah & Asija (1956)* |
| Available phosphorus (kg ha$^{-1}$) | 18.6 | 19.8 | Olsen's extractant method *Olsen et al. (1954)* |
| Available potassium (kg ha$^{-1}$) | 210.5 | 215.6 | Flame photometer *Jackson (1973)* |

(WPEtc). Irrigation inflow divides rice production. WPTCW = rice yield/rain, irrigation, and other inflows. Rice yield equals WPEtc divided by evapotranspiration. Water productivity indices are calculated from establishing and nutrient source treatments (*Kar et al., 2015*).

## Statistical investigation

The experiment used a factorial randomised block design, and all data were analysed using "analysis of variance" (ANOVA) (*Gomez & Gomez, 1984*). The 'F' test determines treatment relevance (variance ratio). SEm ± was determined in each case. The mean difference was tested using 5% critical difference (CD). NS showed non-significant treatment differences. Crop performance productivity, plant nutrient uptake, and soil fertility status data were noted, evaluated, and tallied after a statistical test to get the right result.

## RESULT AND DISCUSSION

### Rice crop growth performance
#### Plant height (cm)

Plant height is vital for studying crop development and treatment effects. In 2019 and 2020, average plant height increased slowly to 41.3 and 42.0 cm at 30 days after transplanting (DAT), linearly at 60 DAT (76.0 and 76.9 cm), and then at a falling speed (94.2 and 95.8 cm at 90 DAT, 98.2 and 97.8 cm at harvest) (Table 3). On an average of two years study, CT-TPR recorded 2.41% increment in plant height than W Bed-TPR in entire crop growth. In both years, crop establishment and fertiliser management methods affected rice plant height at all growth phases. In the 2019 and 2020 Kharif seasons, conventional

tillage transplanted puddled rice (CE1, CT-TPR) was recorded significantly higher than transplanted wide bed rice (CE2, W Bed-TPR) at 30, 60, 90 DAT, and harvest. Over two years, transplanted wide bed rice (CE2, W Bed-TPR) had the lowest mean plant height at 30 DAT (35.0 and 35.6 cm), 60 DAT (69.3 and 70.3 cm), 90 DAT (87.6 and 88.4 cm) and at harvest (90.0 and 90.5 cm) respectively. At 30 (41.3 and 42.0 cm), 60 (76.0 and 76.9 cm), 90 DAT (94.2 and 95.8 cm), and harvest (98.2 and 97.8), the N6 treatment exceeded the other treatments and was at par with the N3 and N5 treatments (Table 3). N8 and N2 had the highest plant height records in both years of the trial and statistically outperformed the other fertiliser administration regimens. N7, N4, and N9 have comparable plant heights at various crop growth stages. The control treatment N1 no NPK had the lowest mean rice plant height at 30 (28.7 and 29.0 cm), 60 (62.3 and 64.8 cm), 90 DAT (79.0 and 78.1 cm) and harvest (80.9 and 81.8 cm) in both research years. Over the two years, N6 produced the tallest plants in all growth phases, followed by N3 and N5, whereas N1 produced the shortest rice plant height. The N6 at CE1 all-growth stage treatment combination was best rice height in two research years respectively.

### Tillers number $m^{-2}$

Tillers per unit area are important in determining how a treatment affects a crop like basmati rice. The average rice tillers per square metre rose linearly up to 90 DAT, but then significantly dropped due to self-thinning, resource shortages, and intra-plant competition (Table 3). Over the two years of the study, rice tillers number ($m^{-2}$) at different rice development stages fluctuated due to crop setup and fertiliser management practices. The interaction effect of tillers per $m^{-2}$ was unaffected by crop establishment and fertiliser management. Different crop planting and fertiliser application methods affect rice tiller number $m^{-2}$ at different growth stages. Transplanted rice on wide beds produced more tillers after being puddled. Traditional transplanted puddled rice (CE1, CT-TPR) had a greater mean tillers number per square metre than rice transplanted into a wide bed (CE2, W Bed-TPR) at varied growth intervals in the 2019 and 2020 Kharif seasons. Wide bed-transplanted rice (CE2, W Bed-TPR) produced fewer mean tillers per square metre than conventional procedures at various phases of rice growth across the two experimentation years. For fertiliser management, tillers number $m^{-2}$ at 30, 60 DAT, and harvest ranged from 228 to 289 and 234 to 294, 251 to 439 and 254 to 443, and 219 to 397 and 220 to 398 for the two years (Table 3). N6 generated the most tillers per square metre at 30, 60, 90, and harvest in both years, outperforming all other treatments except N3 at 30DAT and N3 and N5 at 60, 90, and harvest. N8 and N2 had more tillers per square metre than the other fertiliser application methods in both years of the experiment. Additionally, N7, N4, and N9 had a similar number of tillers per square metre and were comparable. The untreated control plot N1 had the fewest tillers per square metre during growth. Tiller number was unaffected by crop planting techniques and rice nutrition management measures ($m^{-2}$).

### Production of dry matter (g $m^{-2}$)

The strongest indication of crop development that points to the buildup of dry matter is the result of photosynthesizing residue remaining after respiration. The CT-TPR (puddled rice planted using a typical till, (CE1) accumulated much more dry matter gm/m2 than the wide

Chandra et al. (2023), *PeerJ*, DOI 10.7717/peerj.16271

**Table 3** Rice plant height and number of tillers m$^{-2}$ at various development intervals as a result of various crop planting techniques and fertiliser management strategies.

| Treatment | Rice Crop height (cm) | | | | | | | | Tillers number m$^{-2}$ | | | | | | | |
|---|---|---|---|---|---|---|---|---|---|---|---|---|---|---|---|---|
| | 30 DAT | | 60 DAT | | 90 DAT | | At harvest | | 30 DAT | | 60 DAT | | 90 DAT | | At harvest | |
| | 2019 | 2020 | 2019 | 2020 | 2019 | 2020 | 2019 | 2020 | 2019 | 2020 | 2019 | 2020 | 2019 | 2020 | 2019 | 2020 |
| **Crop planting techniques (A)** | | | | | | | | | | | | | | | | |
| CE1 | 36.6 | 37.2 | 70.9 | 72.0 | 89.4 | 90.2 | 91.7 | 92.4 | 264 | 267 | 402 | 406 | 374 | 375 | 359 | 361 |
| CE2 | 35.0 | 35.6 | 69.3 | 70.3 | 87.6 | 88.4 | 90.0 | 90.5 | 269 | 271 | 408 | 412 | 380 | 381 | 364 | 366 |
| SEM ± | 0.48 | 0.49 | 0.56 | 0.58 | 0.59 | 0.62 | 0.62 | 0.63 | 1.79 | 1.80 | 2.33 | 2.37 | 2.10 | 2.13 | 1.95 | 2.05 |
| CD ($p = 0.05$) | 1.37 | 1.41 | 1.62 | 1.66 | 1.70 | 1.79 | 1.77 | 1.82 | 5.15 | 5.17 | 6.70 | 6.80 | 6.03 | 6.13 | 5.62 | 5.89 |
| **Fertiliser management strategies (B)** | | | | | | | | | | | | | | | | |
| N1 | 28.7 | 29.0 | 62.3 | 64.8 | 79.0 | 78.1 | 80.9 | 81.8 | 228 | 234 | 251 | 254 | 235 | 232 | 219 | 220 |
| N2 | 35.6 | 36.2 | 71.1 | 71.6 | 90.0 | 90.3 | 91.1 | 92.4 | 269 | 268 | 420 | 426 | 391 | 392 | 374 | 373 |
| N3 | 40.0 | 40.9 | 74.1 | 75.6 | 93.0 | 93.2 | 96.2 | 96.9 | 284 | 288 | 434 | 437 | 407 | 408 | 392 | 393 |
| N4 | 33.1 | 34.1 | 67.1 | 68.3 | 85.7 | 87.1 | 88.0 | 88.4 | 262 | 261 | 415 | 421 | 383 | 385 | 368 | 371 |
| N5 | 39.1 | 39.4 | 73.8 | 74.5 | 92.2 | 92.6 | 95.3 | 95.9 | 276 | 279 | 429 | 431 | 402 | 404 | 386 | 388 |
| N6 | 41.3 | 42.0 | 76.0 | 76.9 | 94.2 | 95.8 | 98.2 | 97.8 | 289 | 294 | 439 | 443 | 412 | 414 | 397 | 398 |
| N7 | 34.6 | 35.2 | 68.1 | 69.2 | 87.4 | 88.9 | 88.6 | 90.1 | 264 | 265 | 417 | 422 | 385 | 386 | 370 | 371 |
| N8 | 36.7 | 37.5 | 72.4 | 73.1 | 90.8 | 91.3 | 93.2 | 93.6 | 270 | 275 | 426 | 430 | 397 | 398 | 379 | 384 |
| N9 | 32.9 | 33.4 | 65.9 | 66.5 | 84.0 | 86.2 | 86.3 | 86.1 | 258 | 257 | 415 | 418 | 379 | 384 | 365 | 371 |
| SEM ± | 1.01 | 1.04 | 1.19 | 1.23 | 1.26 | 1.32 | 1.31 | 1.35 | 3.80 | 3.81 | 4.94 | 5.02 | 4.45 | 4.53 | 4.15 | 4.35 |
| CD (p =0.05) | 2.91 | 2.98 | 3.43 | 3.52 | 3.61 | 3.79 | 3.76 | 3.87 | 10.93 | 10.96 | 14.20 | 14.43 | 12.80 | 13.01 | 11.92 | 12.50 |
| **Interaction (A X B)** | | | | | | | | | | | | | | | | |
| SEM ± | 1.43 | 1.47 | 1.69 | 1.73 | 1.78 | 1.86 | 1.85 | 1.90 | 5.38 | 5.39 | 6.99 | 7.10 | 6.30 | 6.38 | 5.88 | 6.16 |
| CD ($p = 0.05$) | NS | NS | NS | NS | NS | NS | NS | NS | NS | NS | NS | NS | NS | NS | NS | NS |

Notes.

Crop planting techniques *i.e.*, $E_1$-Conventional puddled transplanted rice (CT- TPR), $E_2$- Wide bed Transplanted rice (W Bed-TPR); Fertilizer management strategies *i.e.*, $N_1$- Control, $N_2$- 100% RDF + $ZnSO_4$ 25 kg ha$^{-1}$, $N_3$- 125% RDN + $ZnSO_4$ 25 kg ha$^{-1}$, $N_4$- STCR based NPK application + $ZnSO_4$ 25 kg ha$^{-1}$, $N_5$- $N_2$+ FYM (5 t ha$^{-1}$), $N_6$- $N_2$+ FYM (5 t ha$^{-1}$) + PSB (5 kg ha$^{-1}$)+ Azotobactor 20 kg ha$^{-1}$, $N_7$- 75% RDN + FYM (5 t ha$^{-1}$) + $ZnSO_4$ 25 kg ha$^{-1}$, $N_8$-75% RDN + FYM (5 t ha$^{-1}$) + PSB (5 kg ha$^{-1}$) + Azotobactor 20 kg ha$^{-1}$ + ZnSO4 25 kg ha$^{-1}$, $N_9$- Organics Practices @ FYM (30 t ha$^{-1}$)+PSB (5 kg ha$^{-1}$) + Azotobactor 20 kg ha$^{-1}$ + $ZnSO_4$ 25 kg ha$^{-1}$, *NS- Non-significant, *DAT- days after transplanting.
raised beds used to transplant rice, (CE2) among the various tillage crop establishment approaches (Table 4). At 60, 90, and harvest, CE1 accumulated substantially more dry matter than the broad bed. Rice transplanted on a wide bed (CE2, W Bed-TPR) reduced dry matter per square metre compared to traditional procedures throughout the two-year experimental trial (Table 4). The N6 treatment's fertiliser management technique produced more dry matter than the other treatments and was comparable to the N3 and N5 treatments at 30, 60, 90, and harvest. N8 and N2 therapies improved dry matter accumulation and were proportionally more effective than dietary management measures in all study years. N7, N4, and N9 showed similar dry matter accumulation tendencies. Both years, control plots had the least dry matter at harvest, 30, 60, and 90 DAS. The trend was consistent throughout.

### Leaf area index

The fast-developing sink's total leaf area per unit ground area is a vital indicator of the plant's total supply for photosynthetic activity. Leaf area index is a significant physiological component that affects crop yield because it affects crop canopy light absorption. The growth metric leaf area index represents the crop's photosynthesizing surface. Fertilizer management and crop establishment affected this parameter. Leaf area index increases till the panicle start (60 days after transplanting) and then decreases at 90 DAT during *Kharif* 2019 and 2020 (Table 4). Wide bed transplanted rice (CE2, W Bed-TPR) had a higher leaf area index (2.29 to 2.31, 3.58 to 3.59, and 3.49 to 3.50) than conventional puddled transplanted rice (CE1, CT-TPR) at 30, 60, and 90 DAT during Kharif 2019 and 2020. Wide bed transplanted rice (CE2, W Bed-TPR) had significantly lower leaf area index at 30, 60, and 90 DAT throughout the two-year research. (2.25 to 2.28, 3.52 to 3.54, and 3.42 to 3.42). Rice nutrition management affected the leaf area index during crop growth. N6 had the highest leaf area index (2.66 to 2.67, 3.88 to 3.89, and 3.79 to 3.81) of the fertiliser management practices at 30, 60, and 90 DAT. However, during both trial years, N3, N5, N8, and N2 outperformed other nutrient techniques proportionally and had greater leaf area indices. The soil leaf area indices of N7, N4, and N9 were equivalent. The control conditions (1.83 to 1.87, 3.18 to 3.22, and 3.07 to 3.08) had the lowest leaf area index during the two years. Fertiliser management and establishment procedures have no interaction effect.

## Physiological studies
### Rate of crop growth (g m$^{-2}$ day$^{-1}$)

The crop growth rate is the most essential growth function because it indicates dry matter outcome per component surface area throughout time. The average crop growth rate (CGR) climbed proportionally between 30 and 60 DAT, dipped progressively between 60 and 90 DAT, and then decreased dramatically as harvest approached in Kharif 2019 and 2020. During both years of study, crop growth (g m$^{-2}$ day$^{-1}$) was similar among crop planting strategies at 0–30 DAT, 60–90 DAT, and 90 DAT-harvest (Table 5). Different fertiliser management methods affected rice crop growth (g m$^{-2}$ day$^{-1}$) at 30-day intervals. Different fertiliser sources produced rice crop growth rates of 9.1 to 13.8 and 9.4 to 13.6, 6.4 to 8.6 and 6.0 to 8.6, 2.5 to 5.4 and 3.1 to 5.8 g m$^{-2}$ day$^{-1}$ at 30 to 60 DAT, 60 to 90 DAT,

Chandra et al. (2023), PeerJ, DOI 10.7717/peerj.16271

**Table 4** Rice plant dry matter accumulation (gm/m$^2$) and leaf area index at various development intervals as a result of various crop planting techniques and fertiliser management strategies.

| Treatment | Dry matter accumulation (gm m$^{-2}$) | | | | | | | | Leaf area index (LAI) | | | | | |
|---|---|---|---|---|---|---|---|---|---|---|---|---|---|---|
| | 30 DAT | | 60 DAT | | 90 DAT | | At harvest | | 30 DAT | | 60 DAT | | 90 DAT | |
| | 2019 | 2020 | 2019 | 2020 | 2019 | 2020 | 2019 | 2020 | 2019 | 2020 | 2019 | 2020 | 2019 | 2020 |
| Crop planting techniques (A) | | | | | | | | | | | | | | |
| CE1 | 245.1 | 249.7 | 603.3 | 612.7 | 869.9 | 875.6 | 1006.0 | 1021.9 | 2.25 | 2.28 | 3.52 | 3.54 | 3.42 | 3.45 |
| CE2 | 235.3 | 240.4 | 582.5 | 587.9 | 830.8 | 834.2 | 958.4 | 967.1 | 2.29 | 2.31 | 3.58 | 3.59 | 3.49 | 3.50 |
| SEM ± | 2.46 | 2.50 | 6.65 | 7.25 | 13.14 | 13.55 | 12.40 | 12.68 | 0.01 | 0.01 | 0.01 | 0.01 | 0.01 | 0.01 |
| CD ($p = 0.05$) | 7.08 | 7.18 | 19.12 | 20.85 | 37.77 | 38.93 | 35.64 | 36.44 | 0.03 | 0.03 | 0.02 | 0.02 | 0.03 | 0.03 |
| Fertiliser management strategies (B) | | | | | | | | | | | | | | |
| N1 | 157.6 | 162.8 | 430.4 | 446.0 | 623.4 | 627.3 | 699.4 | 719.3 | 1.83 | 1.87 | 3.18 | 3.22 | 3.07 | 3.08 |
| N2 | 248.3 | 252.9 | 642.5 | 647.5 | 865.4 | 869.4 | 1,028.9 | 1,043.4 | 2.21 | 2.25 | 3.54 | 3.55 | 3.44 | 3.47 |
| N3 | 265.8 | 272.3 | 654.7 | 666.8 | 917.6 | 929.2 | 1,048.6 | 1,061.4 | 2.55 | 2.59 | 3.81 | 3.83 | 3.72 | 3.74 |
| N4 | 234.7 | 237.3 | 540.9 | 541.6 | 823.7 | 826.0 | 967.1 | 977.9 | 2.13 | 2.16 | 3.39 | 3.40 | 3.27 | 3.30 |
| N5 | 260.5 | 265.2 | 649.0 | 665.0 | 913.6 | 922.4 | 1,048.1 | 1,058.5 | 2.47 | 2.48 | 3.73 | 3.75 | 3.66 | 3.68 |
| N6 | 273.1 | 281.5 | 686.7 | 688.8 | 945.7 | 947.0 | 1,074.8 | 1,085.2 | 2.66 | 2.67 | 3.88 | 3.89 | 3.79 | 3.81 |
| N7 | 235.4 | 239.0 | 553.3 | 556.1 | 839.0 | 841.8 | 976.4 | 988.3 | 2.16 | 2.17 | 3.42 | 3.43 | 3.33 | 3.34 |
| N8 | 254.5 | 260.0 | 644.4 | 652.3 | 908.5 | 910.7 | 1,044.5 | 1,054.8 | 2.32 | 2.33 | 3.62 | 3.64 | 3.55 | 3.57 |
| N9 | 231.9 | 234.4 | 534.2 | 538.6 | 816.2 | 820.2 | 951.9 | 961.6 | 2.12 | 2.13 | 3.37 | 3.38 | 3.25 | 3.27 |
| SEM ± | 5.23 | 5.30 | 14.11 | 15.39 | 27.88 | 28.73 | 26.30 | 26.89 | 0.02 | 0.02 | 0.02 | 0.02 | 0.02 | 0.02 |
| CD (p =0.05) | 15.02 | 15.23 | 40.55 | 44.23 | 80.13 | 82.59 | 75.60 | 77.29 | 0.06 | 0.07 | 0.05 | 0.05 | 0.06 | 0.07 |
| Interaction (A X B) | | | | | | | | | | | | | | |
| SEM ± | 7.39 | 7.49 | 19.95 | 21.76 | 39.43 | 40.63 | 37.19 | 38.03 | 0.029 | 0.034 | 0.023 | 0.026 | 0.030 | 0.034 |
| CD (p =0.05) | NS | NS | NS | NS | NS | NS | NS | NS | NS | NS | NS | NS | NS | NS |

Notes.

Crop planting techniques *i.e.*, CE$_1$-Conventional puddled transplanted rice (CT- TPR), CE$_2$- Wide bed Transplanted rice (W Bed-TPR) ; Fertilizer management strategies *i.e.*, N$_1$- Control, N$_2$- 100% RDF + ZnSO$_4$ 25 kg ha$^{-1}$, N$_3$- 125% RDN + ZnSO$_4$ 25 kg ha$^{-1}$, N$_4$- STCR based NPK application + ZnSO$_4$ 25 kg ha$^{-1}$, N$_5$- N$_2$+ FYM (5 t ha$^{-1}$), N$_6$- N$_2$+ FYM (5 t ha$^{-1}$) + PSB (5 kg ha$^{-1}$)+ Azotobactor 20 kg ha$^{-1}$, N$_7$- 75% RDN + FYM (5 t ha$^{-1}$) + ZnSO$_4$ 25 kg ha$^{-1}$, N$_8$-75% RDN + FYM (5 t ha$^{-1}$) + PSB (5 kg ha$^{-1}$) + Azotobactor 20 kg ha$^{-1}$ + ZnSO4 25 kg ha$^{-1}$, N$_9$- Organics Practices @ FYM (30 t ha$^{-1}$)+PSB (5 kg ha$^{-1}$) + Azotobactor 20 kg ha$^{-1}$ + ZnSO$_4$ 25 kg ha$^{-1}$, *NS- Non-significant, *DAT- days after transplanting.
**Table 5  Rice crop growth rate and relative growth rate as affected by various planting methods and fertiliser management strategies at different growth stages.**

| Treatment | Rate of Crop growth (g/m$^2$/day) | | | | | | Rate of Relative growth (g/g/day) | | | | | |
|---|---|---|---|---|---|---|---|---|---|---|---|---|
| | 30–60 DAT | | 60–90 DAT | | 90 DAT to Harvest | | 30–60 DAT | | 60–90 DAT | | 90 DAT to Harvest | |
| | 2019 | 2020 | 2019 | 2020 | 2019 | 2020 | 2019 | 2020 | 2019 | 2020 | 2019 | 2020 |
| **Crop planting techniques (A)** | | | | | | | | | | | | |
| CE1 | 11.6 | 11.6 | 8.3 | 8.2 | 4.3 | 4.4 | 0.0302 | 0.0299 | 0.0118 | 0.0117 | 0.0048 | 0.0051 |
| CE2 | 12.0 | 12.1 | 8.9 | 8.8 | 4.5 | 4.9 | 0.0304 | 0.0300 | 0.0122 | 0.0119 | 0.0049 | 0.0052 |
| SEM ± | 0.24 | 0.23 | 0.46 | 0.48 | 0.47 | 0.65 | 0.0005 | 0.0005 | 0.0006 | 0.0006 | 0.0005 | 0.0007 |
| CD ($p=0.05$) | NS | NS | NS | NS | NS | NS | NS | NS | NS | NS | NS | NS |
| **Fertiliser management strategies (B)** | | | | | | | | | | | | |
| N1 | 9.1 | 9.4 | 6.4 | 6.0 | 2.5 | 3.1 | 0.0340 | 0.0340 | 0.0120 | 0.0110 | 0.0040 | 0.0048 |
| N2 | 13.2 | 13.2 | 7.4 | 7.4 | 5.4 | 5.8 | 0.0317 | 0.0313 | 0.0098 | 0.0098 | 0.0058 | 0.0061 |
| N3 | 13.0 | 13.1 | 8.8 | 8.7 | 4.4 | 4.4 | 0.0302 | 0.0298 | 0.0112 | 0.0111 | 0.0045 | 0.0044 |
| N4 | 10.2 | 10.1 | 9.4 | 9.5 | 4.8 | 5.1 | 0.0279 | 0.0275 | 0.0139 | 0.0139 | 0.0055 | 0.0058 |
| N5 | 13.2 | 13.3 | 8.8 | 8.6 | 4.5 | 4.5 | 0.0313 | 0.0307 | 0.0113 | 0.0109 | 0.0046 | 0.0046 |
| N6 | 13.8 | 13.6 | 8.6 | 8.6 | 4.3 | 4.6 | 0.0308 | 0.0299 | 0.0106 | 0.0107 | 0.0043 | 0.0046 |
| N7 | 10.6 | 10.6 | 9.5 | 9.5 | 4.6 | 4.9 | 0.0285 | 0.0280 | 0.0139 | 0.0139 | 0.0051 | 0.0054 |
| N8 | 12.8 | 13.1 | 8.8 | 8.6 | 4.5 | 4.8 | 0.0302 | 0.0307 | 0.0114 | 0.0111 | 0.0047 | 0.0049 |
| N9 | 10.1 | 10.1 | 9.4 | 9.4 | 4.5 | 4.7 | 0.0278 | 0.0277 | 0.0140 | 0.0140 | 0.0052 | 0.0054 |
| SEM ± | 0.51 | 0.48 | 0.97 | 1.02 | 0.99 | 1.38 | 0.0012 | 0.0010 | 0.0012 | 0.0013 | 0.0011 | 0.0015 |
| CD ($p=0.05$) | 1.47 | 1.39 | 2.79 | 2.94 | 2.86 | 3.96 | 0.0033 | 0.0030 | 0.0036 | 0.0038 | 0.0031 | 0.0043 |
| **Interaction (A X B)** | | | | | | | | | | | | |
| SEM ± | 0.72 | 0.68 | 1.37 | 1.45 | 1.41 | 1.95 | 0.002 | 0.001 | 0.002 | 0.002 | 0.002 | 0.002 |
| CD ($p=0.05$) | NS | NS | NS | NS | NS | NS | NS | NS | NS | NS | NS | NS |

**Notes.**

Crop planting techniques *i.e.*, CE$_1$-Conventional puddled transplanted rice (CT- TPR), CE$_2$- Wide bed Transplanted rice (W Bed-TPR) ; Fertilizer management strategies *i.e.*, N$_1$- Control, N$_2$- 100% RDF + ZnSO$_4$ 25 kg ha$^{-1}$, N$_3$- 125% RDN + ZnSO$_4$ 25 kg ha$^{-1}$, N$_4$- STCR based NPK application + ZnSO$_4$ 25 kg ha$^{-1}$, N$_5$- N$_2$+ FYM (5 t ha$^{-1}$), N$_6$- N$_2$+ FYM (5 t ha$^{-1}$) + PSB (5 kg ha$^{-1}$)+ Azotobactor 20 kg ha$^{-1}$), N$_7$- 75% RDN + FYM (5 t ha$^{-1}$) + ZnSO$_4$ 25 kg ha$^{-1}$, N$_8$-75% RDN + FYM (5 t ha$^{-1}$) + PSB (5 kg ha$^{-1}$) + Azotobactor 20 kg ha$^{-1}$ + ZnSO4 25 kg ha$^{-1}$, N$_9$- Organics Practices @ FYM (30 t ha$^{-1}$)+PSB (5 kg ha$^{-1}$) + Azotobactor 20 kg ha$^{-1}$ + ZnSO$_4$ 25 kg ha$^{-1}$, *NS- Non-significant, *DAT- days after transplanting.

and 90 DAT after harvest in 2019 and 2020. In two research years, N6 had the highest crop growth (g m$^{-2}$ day$^{-1}$) at 30–60 DAT, N4 at 60–90 DAT, and N2 at 90–90 DAT to harvest. Control conditions (N2) had significantly lower crop growth in both years of experimental testing.

### Rate of relative growth (g g$^{-1}$ day$^{-1}$)

The rate of relative growth (RGR) quantifies a plant's rate of dry matter accumulation in g g$^{-1}$ day$^{-1}$. RGR peaked between 30–60 DAT and subsequently reduced between 60-90 DAT before declining continuously till crop maturity in both years of research (Table 5). In two research years, rice growth rates under diverse crop planting procedures did not differ at 0 to 30 DAT, 60 to 90 DAT, and 90 DAT to harvest. The RGR found substantial differences in fertiliser management practices in both research periods at all crop development stages. The relative growth rate of rice ranged from 0.0278 to 0.0340 and 0.0275 to 0.0340 at 30–60 DAT, 0.0098 to 0.0139 and 0.0098 to 0.0139 at 60–90 DAT, and 0.0040 to 0.0058 and 0.0046 to 0.0061 g g$^{-1}$ day$^{-1}$ at 90DAT to harvest in 2019 and 2020, respectively.
Tables 3–5 show a slight increase in taller plant, tiller count per square metre, dry matter accumulation, LAI, CGR, and RGR from 2019 to 2020. Weather variables including rainfall, daylight hours, and temperature may have caused this growth (Fig. 1). Many growth indicators increased as the crop progressed, although the early vegetative stage had little effect. The superposition of diverse tillage-cum-crop establishment procedures and cumulative seasonal influence induced growth parameter variance over both years. In both years, conventional puddle transplanted rice plots exhibited higher plant growth in late crop growth. Rice plants had more moisture and nutrients, which increased nutrient uptake and led to higher growth characteristics in CE1 than CE2. Water availability maintained higher turgor potential, which led to longer stomatal openings and faster photosynthesis. This accelerates cell division and expansion, increasing growth (*Midya et al., 2021*; *Bhatt et al., 2021*; *Kumar et al., 2019*) found similar results. Higher nitrogen levels boosted plant height, but further dose increases only slightly increased it. Plant height is genetic and less affected by the environment, but the control plant's (N1) plant height was significantly lower than the average for all treatments, suggesting that the rice plant may have been undernourished due to nutrient availability issues. Compared to treatments N5 or N6 with organic manure and chemical fertilizer, plant growth was better. The control and little nitrogen exhibited less dry matter buildup. Poor growth may be due to a lack of rice crop nutrients. Nitrogen is needed for photosynthesis and tissue growth in chlorophyll, protein, and cellulose. N5 and N6 nitrogen fertiliser increased growth. This shows that the rice plant received nitrogen from organic sources gradually and that it could be available at lower doses than synthetic fertilizer, which is commonly available. These findings confirm (*Goutami et al., 2018*; *Jana, Mondal & Mallick, 2020*; *Nataraja et al., 2021*). Adequate dietary sources promoted post-anthesis dry matter accumulation (DMA) in grain (*Iqbal et al., 2020*; *Wu et al., 2021*). (DMA). Post-anthesis DM contribution to grain and DMA at maturity increase grain yield (*Thakur, Uphoff & Antony, 2010*). Grain yield and DMA were reported to be more significantly correlated than DMR by (*Chen et al., 2014*; *Dixit, Singh & Kumar, 2014*). The strong association between DMA and grain production is likely due to high post-anthesis photosynthetic rates, especially in the middle and later stages of grain filling, which helped dry matter building, grain filling, and grain weight. DMA, CGR, RGR, and LAI were lower at lower nitrogen levels (control) than N6, while nutritional supplies at N5 were equivalent in both research years. The treatments' increased growth may boost plant accessible nutrients, which are crucial for growth. Photosynthesis and rice growth require cellulose proteins. Chlorophyll requires water and nutrients. When nitrogen was given at N6 treatment, growth increased and peaked in the study's nutrient sources. This shows that the crop plant has progressively received nutrients and moisture from the nutritional sources, and their availability may be lower than the needed yet easily available moisture. The combination of FYM, pressmud, and inorganic fertilisers may have released sufficient amounts of nutrients through mineralization, resulting in an acceptable amount of accessible nutrients and a better environment for enhanced nutrient uptake and, subsequently, greater crop growth. The rise in plant height in response to the combined application of organic and chemical fertiliser is most likely owing to increased nitrogen availability, which increased leaf area, leading to higher photo assimilates which resulted in more dry matter accumulation. This

finding are in accordance with (*Dass, Sudhishri & Lenka, 2009*; *Roy et al., 2017*; *Jat & Singh, 2019*).

## Yield parameters

### Number of effective tillers (m$^{-2}$)

Productive tillers m$^{-2}$, or fertile tillers with panicles, affect crop grain production. Crop planting and fertiliser application methods considerably affected harvest tiller yield (m$^{-2}$) (Table 6). However, crop planting methods and fertiliser coping mechanisms did not interact. The mean effective tillers (m$^{-2}$) at harvest were 359–364 and 361–366. In 2019 and 2020, rice transplanted on wide beds (CE2, W Bed-TPR) had higher mean effective tillers (364 & 366 m$^{-2}$) than traditional puddled transplanted rice (CE1, CT-TPR). Traditional puddled transplanted rice (CE1, CT-TPR) had a significantly lower mean productive tiller number (359 & 361 m$^{-2}$) during the two-year trial. When compared to fertiliser management options, the N6 (397 & 398 m$^{-2}$) treatment had the highest mean productive tillers. In 2019 and 2020, treatments N8 and N2 had higher mean productive tillers numbers and outperformed the other fertiliser application treatments. The mean number of producing tillers was also comparable in treatments N7, N4, and N9. Over the two-year observational study, N1 had 219 & 220 m$^{-2}$ fewer effective tillers than the other treatments.

### Panicle length (cm)

Panicle length inversely influences grain yield because spikelets and kernel panicle-1 are connected. Panicle length may estimate cereal grain yield. Table 6 indicated that crop planting and fertiliser application methods varied greatly. However, crop planting methods did not affect fertiliser application. In 2019 and 2020, rice transplanted on wide beds (CE2, W Bed-TPR) had longer maximum mean panicle lengths (23.4 & 24.0 cm) than traditional puddled transplanted rice (CE1, CT-TPR). In both years, transplanted puddled rice on the traditional method (CE1, CT-TPR) had a significantly lower mean panicle length (21.9 & 22.3 cm). The mean panicle length was 18.7–26.0 cm in 2019 and 19.6–26.3 cm in 2020. N6 (26.0 & 26.3 cm) had the greatest maximum mean panicle length, except for N3 (24.9 & 25.2 cm) and N5 (24.8 & 24.6 cm). In 2019 and 2020, the mean panicle length outperformed treatments N8, N2, N7, N4, N9, and N1. The control treatment N1 had the shortest panicle length (18.7 & 19.6 cm) during the two-year trial.

### Filled grains panicle$^{-1}$ number

Grain output is directly affected by panicle$^{-1}$ grains. Crop planting and fertiliser application had no noticeable effect. (Table 6). In 2019 and 2020, transplanted puddled rice using the conventional method (CE1, CT-TPR) had 59 and 61 complete grains panicle$^{-1}$, respectively, compared to CE2, W Bed-TPR. Wide bed rice (CE2, W Bed-TPR) had fewer full grains panicle$^{-1}$ (55 & 57) in both years. Except for N3 (63 & 64), N5 (62 & 63), and N8 (61 & 63) treatments, the fertiliser management practices N6 (66 & 67) treatment had the highest filled grains panicle$^{-1}$ number across two years (2019 and 2020). In 2019 and 2020, the filled grains panicle$^{-1}$ number showed the following trend: N2 >N7 >N4 >N9.

**Table 6 Rice yield and its characteristics are affected by various crop planting methods and fertiliser management strategies at harvest stage.**

| Treatment | Yield attributes | | | | | | | | | | Yield (kg ha⁻¹) | | | | | | Harvest index (%) | |
|---|---|---|---|---|---|---|---|---|---|---|---|---|---|---|---|---|---|---|
| | Effective tillers (m²) | | Panicle length (cm) | | Filled grains | | Unfilled grains | | Test weight (gm) | | Grain | | Straw | | Biological | | | |
| | 2019 | 2020 | 2019 | 2020 | 2019 | 2020 | 2019 | 2020 | 2019 | 2020 | 2019 | 2020 | 2019 | 2020 | 2019 | 2020 | 2019 | 2020 |
| **Crop planting techniques (A)** | | | | | | | | | | | | | | | | | | |
| CE1 | 359 | 361 | 21.9 | 22.3 | 59 | 61 | 22 | 20 | 23.0 | 23.9 | 4,043 | 4,129 | 6,271 | 6,379 | 10,315 | 10,508 | 39.08 | 39.13 |
| CE2 | 364 | 366 | 23.4 | 24.0 | 55 | 57 | 25 | 24 | 22.2 | 23.2 | 3,869 | 3,968 | 6,097 | 6,210 | 9,966 | 10,178 | 38.64 | 38.86 |
| SEM ± | 1.95 | 2.05 | 0.48 | 0.51 | 1.2 | 1.2 | 0.5 | 0.5 | 0.22 | 0.23 | 44.2 | 49.1 | 51.7 | 64.6 | 79.1 | 88.2 | 0.35 | 0.36 |
| CD ($p = 0.05$) | 5.62 | 5.89 | 1.37 | 1.46 | 3.3 | 3.5 | 1.3 | 1.5 | NS | NS | 127.0 | 141.0 | 148.7 | 185.8 | 227.4 | 253.4 | NS | NS |
| **Fertiliser management strategies (B)** | | | | | | | | | | | | | | | | | | |
| N1 | 219 | 220 | 18.7 | 19.6 | 43 | 46 | 31 | 31 | 18.5 | 20.1 | 2,273 | 2,351 | 3,722 | 3,872 | 5,994 | 6,223 | 37.69 | 37.72 |
| N2 | 374 | 373 | 22.5 | 23.0 | 58 | 60 | 24 | 23 | 23.2 | 23.9 | 3,955 | 4,034 | 6,368 | 6,438 | 10,323 | 10,473 | 38.29 | 38.47 |
| N3 | 392 | 393 | 24.9 | 25.2 | 63 | 64 | 20 | 16 | 23.4 | 24.5 | 4,689 | 4,793 | 6,932 | 6,977 | 11,621 | 11,769 | 40.36 | 40.75 |
| N4 | 368 | 371 | 21.6 | 22.5 | 54 | 55 | 27 | 27 | 22.4 | 23.4 | 3,752 | 3,782 | 6,112 | 6,153 | 9,863 | 9,935 | 38.04 | 38.06 |
| N5 | 386 | 388 | 24.8 | 24.6 | 62 | 63 | 20 | 19 | 23.2 | 24.3 | 4,375 | 4,492 | 6,725 | 6,835 | 11,100 | 11,327 | 39.35 | 39.59 |
| N6 | 397 | 398 | 26.0 | 26.3 | 66 | 67 | 11 | 8 | 24.5 | 25.2 | 4,903 | 5,018 | 7,120 | 7,253 | 12,023 | 12,272 | 40.80 | 40.89 |
| N7 | 370 | 371 | 22.1 | 22.8 | 55 | 56 | 24 | 26 | 22.8 | 23.6 | 3,852 | 3,965 | 6,218 | 6,375 | 10,070 | 10,340 | 38.25 | 38.34 |
| N8 | 379 | 384 | 22.6 | 23.2 | 61 | 63 | 23 | 20 | 23.2 | 24.2 | 4,230 | 4,343 | 6,555 | 6,722 | 10,785 | 11,065 | 39.23 | 39.28 |
| N9 | 365 | 371 | 20.8 | 21.3 | 52 | 55 | 28 | 28 | 22.4 | 22.9 | 3,577 | 3,658 | 5,905 | 6,025 | 9,482 | 9,683 | 37.73 | 37.84 |
| SEM ± | 4.15 | 4.35 | 1.01 | 1.08 | 2.5 | 2.6 | 1.0 | 1.1 | 0.48 | 0.49 | 93.8 | 104.1 | 109.8 | 137.1 | 167.8 | 187.0 | 0.75 | 0.76 |
| CD ($p = 0.05$) | 11.92 | 12.50 | 2.90 | 3.09 | 7.1 | 7.4 | 2.8 | 3.1 | 1.37 | 1.42 | 269.4 | 299.2 | 315.5 | 394.0 | 482.4 | 537.4 | 2.15 | 2.18 |
| **Interaction (A X B)** | | | | | | | | | | | | | | | | | | |
| SEM ± | 5.69 | 5.97 | 1.43 | 1.52 | 3.5 | 3.6 | 1.4 | 1.5 | 0.67 | 0.70 | 132.6 | 147.2 | 155.3 | 193.9 | 237.4 | 264.5 | 1.06 | 1.07 |
| CD ($p = 0.05$) | NS | NS | NS | NS | NS | NS | NS | NS | NS | NS | NS | NS | NS | NS | NS | NS | NS | NS |

**Notes.**

Crop planting techniques *i.e.*, CE₁-Conventional puddled transplanted rice (CT- TPR), CE₂- Wide bed Transplanted rice (W Bed-TPR) ; Fertilizer management strategies *i.e.*, N₁- Control, N₂- 100% RDF + ZnSO₄ 25 kg ha⁻¹, N₃- 125% RDN + ZnSO₄ 25 kg ha⁻¹, N₄- STCR based NPK application + ZnSO₄ 25 kg ha⁻¹, N₅- N₂+ FYM (5 t ha⁻¹), N₆- N₂+ FYM (5 t ha⁻¹) + PSB (5 kg ha⁻¹)+ Azotobactor 20 kg ha⁻¹, N₇- 75% RDN + FYM (5 t ha⁻¹) + ZnSO₄ 25 kg ha⁻¹, N₈-75% RDN + FYM (5 t ha⁻¹) + PSB (5 kg ha ⁻¹) + Azotobactor 20 kg ha⁻¹ + ZnSO4 25 kg ha⁻¹, N₉- Organics Practices @ FYM (30 t ha⁻¹)+PSB (5 kg ha⁻¹) + Azotobactor 20 kg ha⁻¹ + ZnSO₄ 25 kg ha⁻¹, *NS- Non-significant.

Untreated control N1 (43 & 46) had less full grains panicle$^{-1}$ than the other treatments in both study years.

### Unfilled grains panicle$^{-1}$ number

Although crop planting techniques and fertiliser management practices did not significantly affect rice unfilled grains, they did affect the number of unfilled grains panicles$^{-1}$ (Table 6). In 2019 and 2020, rice transplanted on wide beds (CE2, W Bed-TPR) had 21 and 20 empty grains panicle$^{-1}$, respectively, compared to CE1, CT-TPR. Traditional puddled transplanted rice (CE1, CT-TPR) generated 25 and 24 empty grains panicle$^{-1}$ during the two-year trial. In 2019 and 2020, the untreated N1 control treatment had a significantly higher unfilled grains count panicle$^{-1}$ than the other fertiliser management approaches. However, the unfilled grains count panicle$^{-1}$ showed the following trend: N9>N4 >N7 >N2 >N8>N5 >N3. Treatment N6 (10 & 8) had significantly fewer whole grains panicle$^{-1}$ than the other treatments over both research periods.

### Test (1,000 grain) weight

The grain weight, determined from the test weight of 1,000 grains, is a critical yield metric that demonstrates how well the grain filling operation was done. Average test weights ranged from 18.5 to 24.5 and 20.1 to 25.2 g depending on fertiliser management strategy (Table 6). Multiple crop establishment procedures and interactions between crop planting techniques and fertiliser management strategies did not affect the 1,000 rice seeds' test weight, which is genetically inherited. The N6 treatment had the highest test weights (24.5 and 25.2 g) in fertiliser management strategies, except for N3 in 2019 and N3, N5, and N8 in 2020. In 2019 and 2020, the test weights were N8 = N5 = N2 >N7 >N4 >N9, showing that one therapy was better than the others. Control treatment N1 (18.5 & 20.1) had a much lower test weight than the others in both years.

Yield combines growth and yield attributes. Integrated crop tillage and nutrient methods increased grain and straw yield. CE1 (CT-TPR) produced the most grain cum straw, whereas CE2 produced the least. (Wbed-TPR). N6 treatment increased rice grain and straw yields. enhanced photosynthate translocation and NPK absorption, which speed up photosynthetic product movement from source to sink, and also enhanced production. Improved vegetative development and high yields increased rice grain and straw yields. Higher FYM and bio fertiliser levels affected rice growth, development, productivity, and quality (*Kumar et al., 2019*; *Gautam et al., 2012*; *Daniela, Mark & Bruce, 2017*).

## Yield

Different crop planting and fertiliser management tactics affected the rice harvest index, straw yield, and grain production (Table 6). However, crop planting and fertiliser management had little effect.

### Grain yield (kg ha$^{-1}$)

Rice yield was affected by crop establishment and fertiliser management methods. In 2019 and 2020, conventionally transplanted puddled rice (CE1, CT-TPR) had higher grain yields (4,043 & 4,129 kg ha$^{-1}$) than wide bed-transplanted rice (CE2, W Bed-TPR). Rice

transplanted on wide beds (CE2, W Bed-TPR) produced 3,869 and 3,968 kg ha$^{-1}$ less grain in both years. N6 (4,903 and 5,018 kg ha$^{-1}$) fertiliser management yielded much more grain than the other treatments, which were at parity with N3 (4,689 and 4,793 kg ha$^{-1}$). However, N3, N5, N8, and N2 had statistically better grain yields than the other fertiliser application regimens in both years. Treatments N7, N4, and N9 had comparable grain yield rates. Untreated N1 control treatment (2,273 & 2,351 kg ha$^{-1}$) produced considerably less grain during both research years.

### Straw yield (kg ha$^{-1}$)

Traditional transplanted puddled rice (CE1, CT-TPR) yielded 6,271 and 6,379 kg ha$^{-1}$ in 2019 and 2020, respectively, compared to transplanted rice on a wide bed (CE2, W Bed-TPR). Rice transplanted on a wide bed (CE2, W Bed-TPR) produced the least straw (6,097 & 6,210 kg ha$^{-1}$) of all crop establishment methods in the two trial years. In straw yield, the N6 treatment (7,120 & 7,253 kg ha$^{-1}$) surpassed all other fertiliser management approaches except N3 (6,937 & 6,977 kg ha$^{-1}$). However, throughout both study years, the N3, N5, N8, and N2 treatments produced more straw and statistically outperformed its other fertiliser administration treatments. Treatments N7, N4, and N9 had comparable grain yield rates. The control treatment N1 (3,722 & 3,872 kg ha$^{-1}$) yielded the least straw in both years.

### Biological productivity (kg ha$^{-1}$)

The total grain and straw yields of rice indicate the crop's photosynthetic efficiency and the amount of photosynthetic material left after respiration, which affects agricultural productivity. In 2019 and 2020, transplanted rice on broad bed (CE2, W Bed-TPR) had a lower biological yield (10,315 & 10,508 kg ha$^{-1}$) than transplanted puddled rice in conventionally (CE1, CT-TPR). Over two years, rice transplanted on wide beds (CE2, W Bed-TPR) yielded 9,966 and 10,178 kg ha$^{-1}$ less biologically. N6 treatment had a significantly higher biological output than the treatment alternatives, which were on par with N3 treatment (11,621 & 11,769 kg ha$^{-1}$) among fertiliser management approaches. (12,023 & 12,272 kg ha$^{-1}$). However, in 2019 and 2020, treatment methods N3, N5, N8, and N2 provided a greater biological yield and outperformed the remaining nutrition management treatments statistically. N7, N4, and N9 have equivalent biological yields. Control treatment N1 (5,994 & 6,223 kg ha$^{-1}$) had a far lower biological yield than the other treatments over the two-year trial.

### Harvest index

The harvest index of rice ranged from 37.69 to 37.72 and 40.80 to 40.89 percent among different nutrient sources. Different crop setup processes had similar harvest indexes. The N6 fertiliser management strategy had the highest harvest index (40.80 to 40.89%) in 2019 and 2020, followed by N3 (40.36 & 40.75%) and N5 (39.35 & 39.59%). In both testing years, N3 and N8, which were objectively better fertiliser application treatments, had the highest harvest index. Harvest indices for N7, N2, N4, and N9 are comparable. Untreated control N1 (37.69–37.72 q ha$^{-1}$) had the lowest harvest index during the two research periods.

Genetic potential and crop environment affect rice grain output (*Daniela, Mark & Bruce, 2017*). To maximise yield, a genetically modified crop might be adjusted agronomically. The crop season's superior weather—increased rainfall, temperatures, and sunshine hours—may have contributed to 2020's somewhat higher grain, straw, and biological product yields (Fig. 1). Tillage increased grain, straw, and biological yield. Wide raised beds and traditional puddles for transplanted rice met irrigation and fertiliser scheduling and provided a very responsive supply of dry matter per rice yield. The higher growth may have caused the higher panicle length, effective tillers number of $m^{-2}$, grains $panicle^{-1}$ number, and test grain weight that increased grain production. Conventionally puddled transplanted rice produced 4.2% more grain during the trial. Grain $panicle^{-1}$ increased 3.8% during the experiment. Similarly, test weight rose 3.3% during experimentation. In the bed sowing rice method, water moves from furrow to up bed, increasing crop yields due to increased nutrient delivery and uptake by crop compared to the flat conventional technique. It is inferred that optimal fertiliser can has the capacity to increase yield and, as a result, minimise WFP of rice production during tillage crop established procedures. This findings are in harmony with (*Daniela, Mark & Bruce, 2017*; *Sandhu et al., 2012*; *Naresh et al., 2014*; *Jat et al., 2014*).

## Irrigation investigation
### Total irrigation water consumption and its water productivity
Percolation water per unit of production for low-tillage crop planting and nitrogen control. The untreated control conditions and synthetic fertiliser application yielded the highest yield ($m^3 \ t^{-1}$) per unit of overall percolation water, with values underneath the N1 and N2 treatments of 1,504.2, 1,470.3, and 1,240.3, 1,248.8 $m^3 \ t^{-1}$, respectively (Table 7). In contrast, organics practices N9, N6, and N8 achieved the minimal cumulative percolation water per unit of output ($m^3 \ t^{-1}$) of 884.9, 862.2, 1,035.4, 1,009.3, and 1,046.4, 1,027.7 in 2019 and 2020. Table 7 shows production efficiency for every unit of irrigation (WPIRRI), fully disgusting or cumulative crop water demand (WPTCW), and evapotranspiration (WPETC) based on tillage crop setup and fertiliser management. Crop establishment WPIRRI was highest with CE2 (0.44 kg $m^{-3}$) and CE1 (0.375 kg $m^{-3}$) tillage practices. Though CE2 output was 9.5% lower than CE1, water productivity per irrigation water unit was 17.3% higher due to greater yields with less water. When we compared CE1 and CE2 yields, the difference was scientifically valid, but when we examined the production efficiency for every irrigation water unit, it became evident that CE2 had substantially higher water productivity than CE1. Fertilizer management strategies with varied sources enhanced the WPIRRI with values of 0.20, 0.21; 0.37, 0.37; 0.37, 0.36; and 0.44, 0.44 kg $m^{-3}$ under N1, N2, N4, and N5. In terms of total crop water requirement, CE1 and CE2 crop establishment treatments produced 0.36, 0.36 and 0.41, 0.42 kg $m^{-3}$, respectively (WPTCW). The CE2 treatment's land design minimised evaporation and percolation, saving water. WPTCW was comparable among tillage crop establishment regimes, even though CE2 yielded less. Despite water's expected productivity being similar to N3, N6, N8, and N9, nitrogen dosages increased WPTCW. Evapotranspiration showed a similar trend for water productivity.

**Table 7  Rice water productivity and total water applied have an interaction effect due to crop planting techniques and fertiliser management practices in *kharif* 2019 and 2020 seasons.**

| Treatments | 2019 | | | | | 2020 | | | | |
|---|---|---|---|---|---|---|---|---|---|---|
| | PERC_V | TWU_V | WP$_{IRRI}$ | WP$_{TCW}$ | WP$_{ETC}$ | PERC_V | TWU_V | WP$_{IRRI}$ | WP$_{TCW}$ | WP$_{ETC}$ |
| | (m3 t−1) | | (kg m−3) | | | (m3 t−1) | | (kg m−3) | | |
| **Crop planting techniques (A)** | | | | | | | | | | |
| CE1 | 1,107.6[b] | 2,971.89[a] | 0.38[b] | 0.36[b] | 0.36[b] | 1,077.8[b] | 2,967.5[a] | 0.37[b] | 0.36[b] | 0.36[b] |
| CE2 | 1,186.0[a] | 2,656.34[b] | 0.44[a] | 0.41[a] | 0.42[a] | 1,153.3[a] | 2,579.3[b] | 0.44[a] | 0.42[a] | 0.42[a] |
| SEM ± | 9.04 | 75.37 | 0.005 | 0.004 | 0.004 | 7.27 | 47.56 | 0.005 | 0.005 | 0.005 |
| CD (p = 0.05) | 25.97 | 216.63 | 0.013 | 0.012 | 0.012 | 20.89 | 136.68 | 0.015 | 0.014 | 0.014 |
| **Fertiliser management practices (B)** | | | | | | | | | | |
| N1 | 1,504.2[a] | 5,289.13[a] | 0.20[e] | 0.19[e] | 0.20[e] | 1,470.3[a] | 5,068.5[a] | 0.21[e] | 0.20[e] | 0.20[e] |
| N2 | 1,240.3[b] | 2,882.28[b] | 0.37[d] | 0.35[d] | 0.35[d] | 1,248.8[b] | 2,891.4[b] | 0.37[d] | 0.35[d] | 0.36[d] |
| N3 | 1,238.1[b] | 2,859.44[bc] | 0.45[b] | 0.43[b] | 0.43[b] | 1,145.2[c] | 2,828.9[b] | 0.45[b] | 0.43[b] | 0.44[b] |
| N4 | 1,151.8[c] | 2,631.02[bcd] | 0.37[d] | 0.35[d] | 0.35[d] | 1,119.0[c] | 2,608.5[bc] | 0.36[d] | 0.35[d] | 0.35[d] |
| N5 | 1,150.2[c] | 2,626.74[bcd] | 0.44[b] | 0.42[b] | 0.42[b] | 1,121.8[c] | 2,618.7[bc] | 0.44[b] | 0.42[b] | 0.43[b] |
| N6 | 1,035.4[d] | 2,327.63[de] | 0.56[a] | 0.52[a] | 0.53[a] | 1,009.3[d] | 2,304.7[d] | 0.55[a] | 0.52[a] | 0.53[a] |
| N7 | 1,069.9[d] | 2,417.56[cd] | 0.41[c] | 0.38[c] | 0.39[c] | 1,042.5[d] | 2,384.4[cd] | 0.40[c] | 0.39[c] | 0.39[c] |
| N8 | 1,046.4[d] | 2,356.51[de] | 0.46[b] | 0.43[b] | 0.44[b] | 1,020.7[d] | 2,321.5[d] | 0.46[b] | 0.44[b] | 0.44[b] |
| N9 | 884.9[e] | 1,936.75[e] | 0.41[c] | 0.39[c] | 0.39[c] | 862.2[e] | 1,934.1[e] | 0.40[c] | 0.39[c] | 0.39[c] |
| SEM ± | 19.17 | 159.89 | 0.010 | 0.009 | 0.009 | 15.42 | 100.88 | 0.011 | 0.010 | 0.010 |
| CD (p = 0.05) | 55.09 | 459.53 | 0.028 | 0.026 | 0.026 | 44.31 | 289.94 | 0.031 | 0.030 | 0.030 |
| **Interaction (A X B)** | | | | | | | | | | |
| SEM ± | 27.12 | 226.12 | 0.013 | 0.014 | 0.013 | 21.80 | 142.67 | 0.015 | 0.015 | 0.015 |
| CD (p = 0.05) | NS | NS | NS | NS | NS | NS | NS | NS | NS | NS |

**Notes.**
Crop planting techniques *i.e.*, CE$_1$-Conventional puddled transplanted rice (CT- TPR), CE$_2$- Wide bed Transplanted rice (W Bed-TPR); Fertilizer management strategies *i.e.*, N$_1$- Control, N$_2$- 100% RDF + ZnSO$_4$ 25 kg ha$^{-1}$, N$_3$- 125% RDN + ZnSO$_4$ 25 kg ha$^{-1}$, N$_4$- STCR based NPK application + ZnSO$_4$ 25 kg ha$^{-1}$, N$_5$- N$_2$+ FYM (5 t ha$^{-1}$), N$_6$- N$_2$+ FYM (5 t ha$^{-1}$) + PSB (5 kg ha$^{-1}$)+ Azotobactor 20 kg ha$^{-1}$, N$_7$- 75% RDN + FYM (5 t ha$^{-1}$) + ZnSO$_4$ 25 kg ha$^{-1}$, N$_8$-75% RDN + FYM (5 t ha$^{-1}$) + PSB (5 kg ha $^{-1}$) + Azotobactor 20 kg ha$^{-1}$ + ZnSO4 25 kg ha$^{-1}$, N$_9$- Organics Practices @ FYM (30 t ha$^{-1}$)+PSB (5 kg ha$^{-1}$) + Azotobactor 20 kg ha$^{-1}$ + ZnSO$_4$ 25 kg ha$^{-1}$, *NS- Non-significant, PERC_V = Percolation water volume, TWU_V= Total water use volume, WP$_{IRRI}$= productivity of water in applied irrigation.
WP$_{TCW}$, productivity of water needed in crop total water and WP$_{ETC}$ = productivity of water in as evapo-transpiration only.

Higher fertiliser doses reduced percolation water volume, while N9 and N6 reduced percolation rates due to shorter standing water periods. These suggest that agrarian management (tillage crop planting and fertiliser tactics) affected irrigation and percolation of freshwater more than the rice crop's agricultural climate. Enhanced agro-management methods can boost output and water productivity. Under tillage crop establishment treatments, optimum fertiliser application may boost yield, reducing water use and rice output. Table 7 demonstrates irrigation productivity per irrigation water unit (WPIRRI), gross crop water demand (WPTCW), and evapotranspiration for tillage crop planting and fertiliser management options. (WPETC). WPIRRI was highest for CE2 (0.44 kg m$^{-3}$) and CE1 (0.375 kg m$^{-3}$) tillage crop planting. CE2 produced 9.5% less output than CE1 but 17.3% more irrigation water per unit due to its ability to produce more yield with less water. Although CE1 and CE2 exhibited statistically significant yield differences, CE2 had significantly higher water productivity per irrigation water unit. Nitrogen practices with N1, N2, N4, and N5 dietary supplies increased WPIRRI. The CE1 and CE2 crop

establishment treatments had water productivity of 0.36, 0.36, and 0.41, 0.42 kg m$^{-3}$ based on total agricultural water usage demand (WPTCW). The CE2 treatment's land design minimised evaporation and percolation, saving water. WPTCW was comparable among tillage crop establishment regimes, even though CE2 yielded less. Despite water productivity equivalent to N3, N6, N8, and N9, the WPTCW increased with nitrogen doses. Evapotranspiration showed a similar trend for water productivity. The crop's WFP was higher when no or low dosages of fertilizers were applied, which could explain the low grain yield observed in nutrient stress plots. WFP was dramatically reduced with increasing nutrient levels from control to RDF+ FYM+15 kg% K sap ha$^{-1}$ due to significant yield enhancement under tillage crop planting practices. Total WFP was, on the other hand, substantially lower with zero tillage and furrow-irrigated raised beds with residue retention than under conventional tillage. This findings are in harmony with (*Pastor et al., 2013*; *Gleeson et al., 2012*; *Keys et al., 2014*; *Kiptala et al., 2014*; *Naresh et al., 2017*).

## Nutrient (NPK) content and uptake

The nutrient uptake content (%) and uptake (kg/ha) in rice grain and straw (N, P, and K) showed significant differences between treatments under integrated crop establishing techniques and fertiliser management strategies, respectively. However, crop planting techniques and fertiliser management practises do not interact significantly (Tables 8, 9 and 10).

### Nitrogen concentrations (%) and uptake (kg ha$^{-1}$)

Grain and straw nitrogen uptake depend on treatment values (Table 8). Crop establishment methods varied greatly between treatments. CE1, CT-TPR surpassed CE2, W Bed-TPR in nitrogen concentrations and uptake in rice grain (1.26 & 1.28% and 51.63 & 53.39 kg ha$^{-1}$) and straw (0.44 & 0.49% and 28.36 & 32.01 kg ha$^{-1}$) during the 2019 and 2020 Kharif seasons. Rice transplanted on wide beds had considerably lower nitrogen concentrations and absorption in grain (1.24 & 1.25% and 48.37 & 50.00 kg ha$^{-1}$) and straw (0.42 & 0.47% and 25.83 & 29.41 kg ha$^{-1}$) during the two experimental years (Table 8). Nutrient management affected rice straw and grain nitrogen concentration and uptake. Table 8 revealed that nutrition management approaches boosted grain and straw nitrogen uptake compared to controls. Over two years and many treatments, rice grain nitrogen uptake ranged from 26.23 to 66.55 and 27.43 to 68.65 kg/ha, while straw uptake ranged from 12.44 to 38.93 and 14.47 to 43.99 kg/ha. In 2019 and 2020, N6 treatment had the highest grain (1.36 & 1.37% and 66.55 & 68.65 kg/ha) and straw (0.55 & 0.61% and 38.93 & 43.99 kg/ha) nitrogen concentration and absorption. The N3, N5, N8, and N2 treatments raised nitrogen concentration and uptake in rice grain and straw more than the other nutrition management treatments in both years of the research. The treatments N7, N4, and N9 had comparable grain and straw content and nitrogen uptake. Rice grain (1.15 & 1.16% and 26.33 & 27.43 kg/ha) and straw (0.33 & 0.37% and 12.44 & 14.47 kg/ha) absorbed significantly less nitrogen under control circumstances in both research years.

**Table 8** Rice grain and straw nitrogen (N) concentration (%) and uptake (kg ha$^{-1}$) as a consequence of various crop planting methods and fertiliser management practices.

| Treatment | Nitrogen content (%) | | | | Nitrogen uptake (kg ha$^{-1}$) | | | |
|---|---|---|---|---|---|---|---|---|
| | Grain | | Straw | | Grain | | Straw | |
| | 2019 | 2020 | 2019 | 2020 | 2019 | 2020 | 2019 | 2020 |
| Crop planting techniques (A) | | | | | | | | |
| CE1 | 1.27 | 1.28 | 0.44 | 0.49 | 51.63 | 53.39 | 28.36 | 32.01 |
| CE2 | 1.24 | 1.25 | 0.42 | 0.46 | 48.37 | 50.00 | 25.83 | 29.41 |
| SEM ± | 0.01 | 0.01 | 0.01 | 0.01 | 0.57 | 0.67 | 0.43 | 0.57 |
| CD ($p = 0.05$) | 0.02 | 0.03 | 0.02 | 0.02 | 1.65 | 1.93 | 1.23 | 1.64 |
| Fertiliser management practices (B) | | | | | | | | |
| N1 | 1.15 | 1.16 | 0.33 | 0.37 | 26.00 | 27.43 | 12.44 | 14.47 |
| N2 | 1.25 | 1.26 | 0.43 | 0.49 | 49.44 | 50.89 | 27.52 | 31.56 |
| N3 | 1.31 | 1.32 | 0.50 | 0.55 | 61.55 | 63.30 | 34.44 | 38.40 |
| N4 | 1.23 | 1.24 | 0.38 | 0.43 | 46.19 | 47.05 | 23.04 | 26.17 |
| N5 | 1.29 | 1.30 | 0.48 | 0.52 | 56.38 | 58.36 | 32.51 | 35.29 |
| N6 | 1.36 | 1.37 | 0.55 | 0.61 | 66.55 | 68.65 | 38.93 | 43.99 |
| N7 | 1.24 | 1.25 | 0.39 | 0.44 | 47.93 | 49.62 | 24.45 | 28.23 |
| N8 | 1.25 | 1.27 | 0.44 | 0.49 | 52.74 | 54.95 | 28.95 | 33.09 |
| N9 | 1.21 | 1.23 | 0.37 | 0.42 | 43.21 | 45.04 | 21.59 | 25.17 |
| SEM ± | 0.02 | 0.02 | 0.01 | 0.01 | 1.22 | 1.42 | 0.91 | 1.21 |
| CD ($p = 0.05$) | 0.05 | 0.06 | 0.03 | 0.04 | 3.50 | 4.09 | 2.62 | 3.49 |
| Interaction (A X B) | | | | | | | | |
| SEM ± | 0.03 | 0.03 | 0.02 | 0.02 | 1.72 | 2.01 | 1.29 | 1.72 |
| CD ($p = 0.05$) | NS | NS | NS | NS | NS | NS | NS | NS |

**Notes.**
Crop planting techniques *i.e.*, CE$_1$-Conventional puddled transplanted rice (CT- TPR), CE$_2$- Wide bed Transplanted rice (W Bed-TPR) ; Fertilizer management strategies *i.e.*, N$_1$- Control, N$_2$- 100% RDF + ZnSO$_4$ 25 kg ha$^{-1}$, N$_3$- 125% RDN + ZnSO$_4$ 25 kg ha$^{-1}$, N$_4$- STCR based NPK application + ZnSO$_4$ 25 kg ha$^{-1}$, N$_5$- N$_2$+ FYM (5 t ha$^{-1}$), N$_6$- N$_2$+ FYM (5 t ha$^{-1}$) + PSB (5 kg ha$^{-1}$)+ Azotobactor 20 kg ha$^{-1}$, N$_7$- 75% RDN + FYM (5 t ha$^{-1}$) + ZnSO$_4$ 25 kg ha$^{-1}$, N$_8$-75% RDN + FYM (5 t ha$^{-1}$) + PSB (5 kg ha$^{-1}$) + Azotobactor 20 kg ha$^{-1}$ + ZnSO4 25 kg ha$^{-1}$, N$_9$- Organics Practices @ FYM (30 t ha$^{-1}$)+PSB (5 kg ha$^{-1}$) + Azotobactor 20 kg ha$^{-1}$ + ZnSO$_4$ 25 kg ha$^{-1}$, *NS- Non-significant.

### Phosphorous concentrations (%) and uptake (kg ha$^{-1}$)

Grain and straw phosphorus concentration and uptake varied greatly among crop establishment methods and fertiliser management practices (Table 9). The crop establishment methods' treatments vary greatly. Throughout the 2019 and 2020 Kharif seasons, the CE1, CT-TPR lowered phosphorus concentration and absorption in grain (0.35 & 0.37% and 14.33 & 15.47 kg ha$^{-1}$) and straw (0.186 & 0.195% and 11.97 & 12.74 kg ha$^{-1}$). Rice transplanted on wide beds had lower grain (0.32 & 0.34% and 12.68 & 13.87 kg ha$^{-1}$) and straw (0.167 & 0.172% and 10.42 & 10.94 kg ha$^{-1}$) phosphorus content and absorption over the two-year trial. Grain and straw phosphorus uptake varied under different fertiliser management regimes. In the two Kharif years of 2019 and 2020, the N6 treatment had the highest phosphorus concentration and accumulation in grain (0.37 & 0.38% and 17.98 & 19.21 kg/ha) and straw (0.203 & 0.211% and 14.42 & 15.25 kg/ha), except for N3 in uptake and N3 and N5 in concentration. However, N8, N2, N7, N4, and N9 had higher phosphorus levels in grain and straw, which were indistinguishable. The statistically superior nutritional methods N5, N8, and N2 boosted rice grain and straw

**Table 9  Rice grain and straw phosphorous (P) concentration (%) and uptake (kg ha⁻¹) in rice grain and straw as a consequence of various crop planting methods and fertiliser management practices.**

| Treatment | Phosphorous content (%) | | | | Phosphorous uptake (kg ha⁻¹) | | | |
|---|---|---|---|---|---|---|---|---|
| | Grain | | Straw | | Grain | | Straw | |
| | 2019 | 2020 | 2019 | 2020 | 2019 | 2020 | 2019 | 2020 |
| Crop planting techniques (A) | | | | | | | | |
| CE1 | 0.35 | 0.37 | 0.186 | 0.195 | 14.33 | 15.47 | 11.97 | 12.74 |
| CE2 | 0.32 | 0.34 | 0.167 | 0.172 | 12.68 | 13.87 | 10.42 | 10.94 |
| SEM ± | 0.01 | 0.01 | 0.002 | 0.003 | 0.22 | 0.25 | 0.16 | 0.19 |
| CD ($p = 0.05$) | 0.02 | 0.02 | 0.007 | 0.007 | 0.63 | 0.71 | 0.45 | 0.55 |
| Fertiliser management practices (B) | | | | | | | | |
| N1 | 0.27 | 0.28 | 0.109 | 0.108 | 6.08 | 6.48 | 4.04 | 4.18 |
| N2 | 0.34 | 0.36 | 0.185 | 0.193 | 13.55 | 14.74 | 11.81 | 12.41 |
| N3 | 0.37 | 0.38 | 0.197 | 0.203 | 17.10 | 18.19 | 13.67 | 14.22 |
| N4 | 0.33 | 0.35 | 0.170 | 0.177 | 12.44 | 13.36 | 10.42 | 10.90 |
| N5 | 0.35 | 0.38 | 0.192 | 0.203 | 15.29 | 16.96 | 12.89 | 13.85 |
| N6 | 0.37 | 0.38 | 0.203 | 0.211 | 17.98 | 19.21 | 14.42 | 15.25 |
| N7 | 0.34 | 0.36 | 0.185 | 0.189 | 13.03 | 14.36 | 11.48 | 12.03 |
| N8 | 0.34 | 0.37 | 0.187 | 0.194 | 14.46 | 15.94 | 12.26 | 13.06 |
| N9 | 0.33 | 0.35 | 0.165 | 0.176 | 11.61 | 12.79 | 9.73 | 10.66 |
| SEM ± | 0.01 | 0.02 | 0.005 | 0.005 | 0.47 | 0.53 | 0.33 | 0.41 |
| CD ($p = 0.05$) | 0.04 | 0.05 | 0.014 | 0.016 | 1.34 | 1.51 | 0.95 | 1.17 |
| Interaction (A X B) | | | | | | | | |
| SEM ± | 0.02 | 0.03 | 0.007 | 0.008 | 0.66 | 0.74 | 0.47 | 0.58 |
| CD ($p = 0.05$) | NS | NS | NS | NS | NS | NS | NS | NS |

**Notes.**
Crop planting techniques *i.e.*, CE₁-Conventional puddled transplanted rice (CT- TPR), CE₂- Wide bed Transplanted rice (W Bed-TPR) ; Fertilizer management strategies *i.e.*, N₁- Control, N₂- 100% RDF + ZnSO₄ 25 kg ha⁻¹, N₃- 125% RDN + ZnSO₄ 25 kg ha⁻¹, N₄- STCR based NPK application + ZnSO₄ 25 kg ha⁻¹, N₅- N₂+ FYM (5 t ha⁻¹), N₆- N₂+ FYM (5 t ha⁻¹) + PSB (5 kg ha⁻¹)+ Azotobactor 20 kg ha⁻¹, N₇- 75% RDN + FYM (5 t ha⁻¹) + ZnSO₄ 25 kg ha⁻¹, N₈-75% RDN + FYM (5 t ha⁻¹) + PSB (5 kg ha⁻¹) + Azotobactor 20 kg ha⁻¹ + ZnSO4 25 kg ha⁻¹, N₉- Organics Practices @ FYM (30 t ha⁻¹)+PSB (5 kg ha⁻¹) + Azotobactor 20 kg ha⁻¹ + ZnSO₄ 25 kg ha⁻¹, *NS- Non-significant.

phosphorus uptake. In grain and straw, N7, N4, and N9 had identical phosphorus content and absorption patterns. Control treatment N1 absorbed less phosphorus in rice grain (0.27 & 0.28% and 6.08 & 6.48 kg/ha) and straw (0.109 & 0.108% and 4.04 & 4.18 kg/ha) than the other treatments over the two years.

### Potassium concentrations (%) and uptake (kg ha⁻¹)

Rice grains and straw absorbed different amounts of potassium (%) under different crop planting and fertiliser management practices (Table 10). Straw and rice grains absorb potassium differently due to agricultural establishment procedures. CE1, CT-TPR showed higher potassium content and absorption than CE2, W Bed-TPR in grain (0.43 & 0.46%, 17.79 & 19.09 kg ha⁻¹) and straw (1.59 & 1.64%, 100.82 & 105.60 kg ha⁻¹). Wide bed transplanted rice absorbed less potassium in grain (0.40 & 0.42% and 15.79 & 17.19 kg ha⁻¹) and straw (1.55 & 1.59% and 95.01 & 99.18 kg ha⁻¹) over two years. Fertilizer management greatly affected rice grain and straw potassium uptake. Except for N3 in Kharif 2020, N6 in 2019 had the highest potassium concentration and accumulation in

**Table 10  Rice grain and straw potassium (K) concentration (%) and uptake (kg ha$^{-1}$) as a consequence of various crop planting methods and fertiliser management practices.**

| Treatment | Potassium content (%) | | | | Potassium uptake (kg ha$^{-1}$) | | | |
|---|---|---|---|---|---|---|---|---|
| | Grain | | Straw | | Grain | | Straw | |
| | 2019 | 2020 | 2019 | 2020 | 2019 | 2020 | 2019 | 2020 |
| Crop planting techniques (A) | | | | | | | | |
| CE1 | 0.43 | 0.46 | 1.59 | 1.64 | 17.79 | 19.09 | 100.82 | 105.60 |
| CE2 | 0.40 | 0.42 | 1.55 | 1.59 | 15.79 | 17.19 | 95.01 | 99.18 |
| SEM ± | 0.005 | 0.006 | 0.01 | 0.02 | 0.26 | 0.33 | 1.20 | 1.59 |
| CD ($p = 0.05$) | 0.01 | 0.02 | 0.04 | 0.05 | 0.73 | 0.94 | 3.46 | 4.57 |
| Fertiliser management practices (B) | | | | | | | | |
| N1 | 0.34 | 0.37 | 1.40 | 1.44 | 7.71 | 8.59 | 51.97 | 56.14 |
| N2 | 0.43 | 0.45 | 1.59 | 1.62 | 17.06 | 18.26 | 101.27 | 104.34 |
| N3 | 0.44 | 0.48 | 1.64 | 1.69 | 20.62 | 23.00 | 113.89 | 117.66 |
| N4 | 0.41 | 0.42 | 1.54 | 1.60 | 15.26 | 15.83 | 94.10 | 98.28 |
| N5 | 0.44 | 0.47 | 1.63 | 1.67 | 19.23 | 21.31 | 109.55 | 113.96 |
| N6 | 0.46 | 0.49 | 1.70 | 1.71 | 22.71 | 24.68 | 120.94 | 124.14 |
| N7 | 0.42 | 0.42 | 1.57 | 1.61 | 16.31 | 16.80 | 97.33 | 102.57 |
| N8 | 0.43 | 0.46 | 1.61 | 1.64 | 18.26 | 19.84 | 105.40 | 110.03 |
| N9 | 0.39 | 0.41 | 1.47 | 1.57 | 13.94 | 14.95 | 86.79 | 94.38 |
| SEM ± | 0.011 | 0.013 | 0.03 | 0.04 | 0.54 | 0.69 | 2.55 | 3.38 |
| CD ($p = 0.05$) | 0.03 | 0.04 | 0.08 | 0.11 | 1.55 | 1.99 | 7.34 | 9.70 |
| Interaction (A X B) | | | | | | | | |
| SEM ± | 0.015 | 0.018 | 0.03 | 0.06 | 0.77 | 0.98 | 3.61 | 4.78 |
| CD ($p = 0.05$) | NS | NS | NS | NS | NS | NS | NS | NS |

**Notes.**
Crop planting techniques *i.e.*, CE$_1$-Conventional puddled transplanted rice (CT- TPR), CE$_2$- Wide bed Transplanted rice (W Bed-TPR) ; Fertilizer management strategies *i.e.*, N$_1$- Control, N$_2$- 100% RDF + ZnSO$_4$ 25 kg ha$^{-1}$, N$_3$- 125% RDN + ZnSO$_4$ 25 kg ha$^{-1}$, N$_4$- STCR based NPK application + ZnSO$_4$ 25 kg ha$^{-1}$, N$_5$- N$_2$+ FYM (5 t ha$^{-1}$), N$_6$- N$_2$+ FYM (5 t ha$^{-1}$) + PSB (5 kg ha$^{-1}$)+ Azotobacter 20 kg ha$^{-1}$, N$_7$- 75% RDN + FYM (5 t ha$^{-1}$) + ZnSO$_4$ 25 kg ha$^{-1}$, N$_8$-75% RDN + FYM (5 t ha$^{-1}$) + PSB (5 kg ha$^{-1}$) + Azotobacter 20 kg ha$^{-1}$ + ZnSO4 25 kg ha$^{-1}$, N$_9$- Organics Practices @ FYM (30 t ha$^{-1}$)+PSB (5 kg ha$^{-1}$) + Azotobacter 20 kg ha$^{-1}$ + ZnSO$_4$ 25 kg ha$^{-1}$, *NS- Non-significant.

straw (1.70 & 1.71% and 120.94 and 124.14 kg/ha) and grain (0.46 & 0.49% and 22.71 and 24.68 kg/ha). Grain and straw have higher phosphorus contents. even though they were comparable. Compared to the other nutrition management treatments, the N5, N8, and N2 treatments boosted rice grain and straw nutrient absorption. N7, N4, and N9 also demonstrated similar potassium uptake in rice grains and straw. Under control conditions (N1), rice grain potassium concentration and absorption were significantly lower (0.34 & 0.37% and 7.71 & 8.59 kg/ha) and straw nitrogen uptake was significantly higher (1.40 & 1.44% and 51.97 & 56.14 kg/ha) for both research years.

Using the nutrients' content in the appropriate part and their production per hectare, rice grain and straw's potassium, nitrogen, and phosphorus consumption was independently measured. Summarizing grain and straw NPK uptake predicted NPK intake. CE2 had the lowest total NPK uptake, while it had the highest grain, straw, and overall NPK intake (Wide bed-TPR). Nutrient methods also affected NPK intake. N6 increases grain and straw NPK content and absorption. Younger seedlings produce more dry matter and tillers,

increasing production and nutrient clearance. This supports *Tomar et al. (2018)* and *Puli et al. (2017)*.

## Post harvest nutrient status of soil

### Available Nitrogen (kg ha$^{-1}$)

Planting methods significantly affected nitrogen availability. Conventional transplanted puddled rice (CE1, CT-TPR) had higher soil nitrogen availability than wide bed transplanted rice (CE2, W Bed-TPR) in Kharif 2019 and 2020. (225.91 & 228.80 kg ha$^{-1}$). Over the two-year experimental study, rice transplanted on wide bed (CE2, W Bed-TPR) had considerably lower soil nitrogen availability (219.18 & 221.86 kg ha$^{-1}$) (Table 11). Fertilizer management greatly affected soil nitrogen availability. The N6 treatment (241.89 & 244.91 kg ha$^{-1}$) raised soil nitrogen more than the other fertiliser management strategies and was comparable to the N3 treatment. N5, N8, and N2 had better soil nitrogen availability and were statistically more effective than all other fertiliser management approaches in both years of the research. N7, N4, and N9 had similar soil nitrogen availability rates. The untreated control N1 treatment (195.56 & 197.90 kg ha$^{-1}$) had significantly less soil nitrogen availability in both research years.

### Available phosphorous (kg ha$^{-1}$)

Planting strategies significantly affected soil phosphorus. Conventionally transplanted pubbled rice (CE1, CT-TPR) had considerably higher soil phosphorus availability (16.52 & 18.41 kg ha$^{-1}$) than wide-bed transplanted rice. (CE2, W Bed-TPR). In two years, transplanted rice on wide bed rice (CE2, W Bed-TPR) had significantly lower soil phosphorus availability. (15.6 & 16.31 kg ha$^{-1}$, respectively). Fertilizer management affected soil phosphorus availability. (Table 11). The N6 fertiliser management treatment (18.73 & 20.32 kg ha$^{-1}$) outperformed the others and was comparable to the N3 and N5 treatments in soil phosphorus availability. In both experiment years, the N8 and N2 treatments had more soil accessible phosphorus than the fertiliser management treatments. The treatments N7, N4, and N9 also increased soil phosphorus availability at the same rate. The untreated control N1 had much less soil phosphorus than the other treatments (10.54 & 12.38 kg ha$^{-1}$).

### Available potassium (kg ha$^{-1}$)

Various planting choices increased soil potassium availability (Table 11). Wide bed transplanted rice (CE2, W Bed-TPR) had considerably lower soil potassium availability (205.07 & 206.35 kg ha$^{-1}$) than puddled rice (CE1, CT-TPR). Transplanted rice on a wide bed (CE2, W Bed-TPR) had considerably decreased soil potassium availability in both years. (200.66 & 202.97 kg ha$^{-1}$). Fertilizer management greatly affected soil potassium availability. Except for N3, N6 has the highest soil potassium availability (216.42 & 219.42 kg ha$^{-1}$). The treatments N5, N8, and N2 had better soil potassium availability and were statistically more effective than the other fertiliser management approaches in both years of the experiment. The soil potassium availability levels of N7, N4, and N9 were also comparable. Over the two-year experimental investigation, the control treatment N1 had less readily available potassium (184.43 & 186.01 kg ha$^{-1}$) in soil than the other treatments.

**Table 11** The impact of interactions between distinct rice crop planting techniques and fertilizer management approaches as it relates to the post-harvest nutrient status of the soil.

| Treatment | Available nutrients (kg ha$^{-1}$) | | | | | | | |
|---|---|---|---|---|---|---|---|---|
| | Nitrogen (N) | | Phosphorus (P$_2$O$_5$) | | Potassium (K$_2$O) | | Organic carbon (%) | |
| | 2019 | 2020 | 2019 | 2020 | 2019 | 2020 | 2019 | 2020 |
| | Crop planting techniques (A) | | | | | | | |
| CE1 | 225.91 | 228.80 | 16.52 | 18.41 | 205.07 | 206.55 | 0.47 | 0.48 |
| CE2 | 219.18 | 221.86 | 15.06 | 16.31 | 200.66 | 202.97 | 0.46 | 0.47 |
| SEM ± | 1.63 | 1.84 | 0.32 | 0.36 | 0.89 | 1.17 | 0.002 | 0.004 |
| CD ($p = 0.05$) | 4.70 | 5.29 | 0.93 | 1.03 | 2.55 | 3.36 | 0.006 | 0.010 |
| | Fertiliser management practices (B) | | | | | | | |
| N1 | 195.56 | 197.90 | 10.54 | 12.38 | 184.43 | 186.01 | 0.40 | 0.41 |
| N2 | 225.19 | 227.79 | 16.76 | 18.30 | 203.67 | 204.98 | 0.48 | 0.48 |
| N3 | 239.71 | 242.48 | 18.26 | 19.87 | 211.65 | 215.94 | 0.50 | 0.51 |
| N4 | 214.75 | 217.48 | 14.39 | 15.97 | 199.08 | 200.95 | 0.44 | 0.45 |
| N5 | 225.58 | 233.25 | 17.21 | 19.10 | 207.38 | 206.88 | 0.49 | 0.50 |
| N6 | 241.89 | 244.91 | 18.73 | 20.32 | 216.42 | 219.42 | 0.51 | 0.52 |
| N7 | 219.33 | 222.10 | 15.68 | 16.91 | 200.68 | 204.21 | 0.45 | 0.46 |
| N8 | 229.44 | 227.87 | 16.98 | 18.35 | 205.75 | 206.06 | 0.49 | 0.50 |
| N9 | 211.47 | 214.20 | 13.61 | 15.07 | 196.73 | 198.45 | 0.44 | 0.43 |
| SEM ± | 3.47 | 3.90 | 0.69 | 0.76 | 1.88 | 2.48 | 0.005 | 0.007 |
| CD ($p = 0.05$) | 9.97 | 11.22 | 1.98 | 2.18 | 5.40 | 7.12 | 0.014 | 0.022 |
| | Interaction (A X B) | | | | | | | |
| SEM ± | 4.90 | 5.52 | 0.97 | 1.07 | 2.66 | 3.50 | 0.007 | 0.011 |
| CD ($p = 0.05$) | NS | NS | NS | NS | NS | NS | NS | NS |

**Notes.**
Crop planting techniques i.e., CE$_1$-Conventional puddled transplanted rice (CT- TPR), CE$_2$- Wide bed Transplanted rice (W Bed-TPR) ; Fertilizer management strategies i.e., N$_1$- Control, N$_2$- 100% RDF + ZnSO$_4$ 25 kg ha$^{-1}$, N$_3$- 125% RDN + ZnSO$_4$ 25 kg ha$^{-1}$, N$_4$- STCR based NPK application + ZnSO$_4$ 25 kg ha$^{-1}$, N$_5$- N$_2$+ FYM (5 t ha$^{-1}$), N$_6$- N$_2$+ FYM (5 t ha$^{-1}$) + PSB (5 kg ha$^{-1}$)+ Azotobacter 20 kg ha$^{-1}$, N$_7$- 75% RDN + FYM (5 t ha$^{-1}$) + ZnSO$_4$ 25 kg ha$^{-1}$, N$_8$-75% RDN + FYM (5 t ha$^{-1}$) + PSB (5 kg ha$^{-1}$) + Azotobacter 20 kg ha$^{-1}$ + ZnSO4 25 kg ha$^{-1}$, N$_9$- Organics Practices @ FYM (30 t ha$^{-1}$)+PSB (5 kg ha$^{-1}$) + Azotobacter 20 kg ha$^{-1}$ + ZnSO$_4$ 25 kg ha$^{-1}$, *NS- Non-significant.

### Organic carbon (%)

Planting practices affected soil organic carbon. Typical transplanted puddled rice (CE1, CT-TPR) had greater organic carbon (0.47 & 0.46%) than wide-bed rice (CE2, W Bed-TPR). Throughout the two experimental periods, transplanted rice on wide bed (CE2, W Bed-TPR) had the lowest organic carbon (0.48 & 0.47%) (Table 11). Fertilizer management affects soil organic carbon. In soil potassium availability, N6 (0.51 & 0.52%) differed significantly from the other treatments, except N3 and N5. N8 and N2 had increased soil-available organic carbon and were statistically better than the other fertiliser management treatments in both years. N7, N4, and N9 also have similar amounts of organic soil carbon. Control treatment N1 (0.40 & 0.41%) had significantly less accessible organic soil carbon than the other treatments in both years.

After continuous application of organic and inorganic sources of nourishment, the CE1 (CT-TPR) plot had the highest soil nutrients (NPK) at harvest compared to the CE2 (Wide bed-TPR) plot. Organic and chemical fertilisers work better together to increase soil fertility and physical condition. In all INM modules, harvest increased soil NPK availability

relative to inorganic fertiliser. The CE1 (CT-TPR) treatment had the highest soil organic carbon after crop harvest. Traditional methods boosted root development, soil nutrient availability and absorption, and nutrient transfer from roots to shoots and grains, which increased growth and yield. N6 (100% RDF + ZnSO4 25 kg ha$^{-1}$ + FYM 5 t ha$^{-1}$ + PSB 5 kg ha$^{-1}$ + Azotobactor 20 kg ha$^{-1}$) increased soil organic carbon because FYM and biofertilizers boost it. N6 (100% RDF + ZnSO4 25 kg ha$^{-1}$ + FYM (5 t ha$^{-1}$) + PSB (5 kg ha$^{-1}$) + Azotobactor (20 kg ha$^{-1}$) caused the greatest pH drop, but INM modules produced neutral soil pH and EC. (N2) (*Dubey, Sharma & Dubey, 2014*; *Bharose et al., 2017*) reported comparable results.

## CONCLUSION

According to a two-year study on tillage, nutrient interaction effects, and basmati rice on the Indo-Gangetic Plain, puddling is the most popular crop establishment method. However, it significantly reduces rice-wheat productivity and sustainability. In western Uttar Pradesh, India, two-year research showed that grain yields can be high without puddling. Planting on large raised beds without puddles may be a viable option for farmers with the right advice. Rice planted on wide raised beds without puddles can offer equivalent yields if weeds are controlled. The best tillage establishment and fertiliser management practices increased rice crop growth, productivity, nutrient uptake, and post-harvest nutrient availability. WBed-TPR plots outperformed CT-TPR plots in crop production, water productivity, nutrient uptake, and soil fertility. According to research, conservation tillage increases rice crop productivity and soil health while helping the environment. Even if traditional fertiliser boosts modern farming, it harms the environment. Fertilizer management using inorganic and organic manure improves rice performance, production, plant and soil health (FYM). Compared to other establishment technologies and nutrient coping strategies, conventionally transplanted puddled rice (CE1, CT-TPR) with N6 improved rice crop yield, concentration, and uptake of NPK and soil health. For long-term rice productivity, this study suggests optimum tillage and fertiliser management. Local governments should help farms employ conservation tillage to optimise tillage and fertiliser use to boost crop growth, soil health, and crop water productivity. Thus, before choosing a management strategy, it is crucial to evaluate plant water and nutrient shortfall yield losses in varied tillage/nutrient sources.

## ACKNOWLEDGEMENTS

The administration of the Sardar Vallabbhai Patel University of Agriculture and Technology in Meerut, Uttar Pradesh, India were very helpful in putting out this experiment, and the authors thank them for their assistance. We also value the technical support.

### Funding

This study was supported by the Researchers Supporting Project number (RSPD2023R686), King Saud University, Riyadh, Saudi Arabia. The funders had no role in study design, data collection and analysis, decision to publish, or preparation of the manuscript.

### Grant Disclosures

The following grant information was disclosed by the authors:
King Saud University, Riyadh, Saudi Arabia:  RSPD2023R686.

### Competing Interests

The authors declare there are no competing interests.

### Author Contributions

- Mandapelli Sharath Chandra conceived and designed the experiments, performed the experiments, analyzed the data, prepared figures and/or tables, authored or reviewed drafts of the article, and approved the final draft.
- R.K. Naresh conceived and designed the experiments, performed the experiments, analyzed the data, prepared figures and/or tables, authored or reviewed drafts of the article, and approved the final draft.
- Rajan Bhatt performed the experiments, analyzed the data, prepared figures and/or tables, authored or reviewed drafts of the article, and approved the final draft.
- Praveen V. Kadam performed the experiments, analyzed the data, authored or reviewed drafts of the article, and approved the final draft.
- Manzer H. Siddiqui conceived and designed the experiments, performed the experiments, authored or reviewed drafts of the article, and approved the final draft.
- Abdel-Rhman Z. Gaafar conceived and designed the experiments, performed the experiments, analyzed the data, prepared figures and/or tables, authored or reviewed drafts of the article, and approved the final draft.
- Md Atikur Rahman performed the experiments, authored or reviewed drafts of the article, and approved the final draft.

### Data Availability

The raw data are available in the Supplemental Files.

### Supplemental Information

Supplemental information for this article can be found online at http://dx.doi.org/10.7717/peerj.16271#supplemental-information.

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
