# Peer review of "Conservation tillage and fertiliser management strategies impact on basmati rice (Oryza sativa L): crop performance, crop water productivity, nutrient uptake and fertility status of the soil under rice-wheat cropping system"

_PeerJ, doi:10.7717/peerj.16271_

## Round 0.1 · original submission · Major Revisions

The authors have been requested to carefully review the reviewers' comments and implement the suggested changes in their manuscript. In addition, they have also been asked to thoroughly proofread their entire manuscript to correct any English typographical errors.

Reviewer 1 ·

Basic reporting

Dear Editor,
I have suggested some corrections in the manuscript.

Experimental design

OK

Validity of the findings

OK

Annotated reviews are not available for download in order to protect the identity of reviewers who chose to remain anonymous.

Reviewer 2 ·

Basic reporting

The study provides important insights into the potential benefits of conservation tillage and integrated nutrient management strategies for enhancing basmati rice productivity and sustainability. I believe that this manuscript would be of interest to your readers.

Experimental design

No comments

Validity of the findings

No comments

Additional comments

Paper is suitable and may be accepted for publication

Reviewer 3 ·

Basic reporting

At present the article is average in language and requires professional English. There are typos and grammatical errors and incomplete sentences throughout the text that requires major revision.The research article titled “Conservation tillage and fertilizer management strategies impacts on basmati rice (Oryza sativa L): Crop performance, crop water productivity, nutrient uptake and fertility status of the soil under Rice- Wheat cropping system” deals with the impact of judicious fertilizer use and conservation tillage on an important cropping pattern Rice-Wheat. The research article is well formulated, relevant and carried out intensively which will generate scientific insight to agronomic technologies under water stress conditions in Rice-wheat cropping system. The study is original fitting the scope of the journal. In my opinion, this review article lacks deep insight explanation overall. The article vocabulary seems average for the standards of the journal. The overall word strength of the article can be worked upon and more fluent language of vocabulary can be served. There are several minor typing errors throughout the text. The result and discussion part consists of only results and there is absence of mechanism and references supporting the claims. Therefore, I would suggest the authors to carry out a careful and extensive revision of the text and include the required mentioned data to make the article more significant and impactful.

Title: Kindly recast the title as the title is not catchy and seems lengthy.
Abstract: Abstract should be recasted with information of the findings in result section and improve the vocabulary. Shorten the keywords.
Introduction: Restructure the introduction part with a conceptual understanding and include more recent references pertaining to conservation agriculture.
Material methods: Exhaustive explanation about soil parameters to be avoided. Include benefit cost ratio of the study.
Result and discussion: Discussion part is missing.


Section Line no Comments
Abstract 20 Recast the line.
23 ‘Change the orientation of the line by substituting the words in capital
32 Quantify ‘better taller plants with increase percentage and higher dry matter. Avoid superficial words like better and higher.
36 N6- Kindly elaborate
38 Mention the ‘method of production’.
50 Recast the line
52 Provide insight into phonological studies.
Introduction 62 Kindly provide reference for the statement.
63 ‘Nation’ provide insight about the nation mentioned in the text
69 Kindly recast with corrected grammar.
73 Elaborate synthetic cost.
77 Complete the sentence “creating imbalance”.
78-79 Reference.
80 Since its introduction. Elaborate ‘its’.
Materials and methods 99 Kindly recast
103 Restructure the sentences correcting the grammar and vocabulary. (0C)
110 ‘PB-1509’- provide additional information about the rice variety .
122 ‘lost plough’
Result and Discussion 157 ‘outgrew’ mention significant or non significant and % increase of height in ‘CT-TPR’
159 Elaborate ‘comparable’
170-171 Contradictory statements
186 Recast the line

Experimental design

The experimental design is unambiguous and relevant. The research is within the scope of the journal and will fulfill the knowledge gap.

Validity of the findings

Findings not assessed with relevant literature.

Additional comments

Discussion is missing and include cost benefit ratio.

---

## Round 0.2 · Minor Revisions

The authors are requested to revise the manuscript as per the suggestions of reviewers.

Reviewer 1 ·

Basic reporting

The result section of abstract is unclear, it should be explained clearly. Some of the necessary corrections suggested in pdf file and highlighted.

Experimental design

The data presented in the table should have interaction CD.

Validity of the findings

The manuscript may be considered for publication after implementation of suggested corrections.

Annotated reviews are not available for download in order to protect the identity of reviewers who chose to remain anonymous.

Reviewer 3 ·

Basic reporting

Thank you very much to the authors for revising the manuscript thoroughly. However, there is room to improve the sentences, data analysis and typographical errors.

Experimental design

NA

Validity of the findings

Data analysis is needed to be revalidated.

Additional comments

NA

---

## Round 0.3 · accepted · Accept

The manuscript is acceptable.

The Section Editor noted:

> The text portion is fine; however, there were a few issues with the accompanying tables which seemed to harbor some un-resolved edits from previous versions.